# 3DGS-HPC: Distractor-free 3D Gaussian Splatting with Hybrid Patch-wise Classification

**Jiahao Chen** [1 2]  **Yipeng Qin** [3]  **Ganlong Zhao** [4]  **Xin Li** [2]  **Wenping Wang** [2]  **Guanbin Li** [1 5]

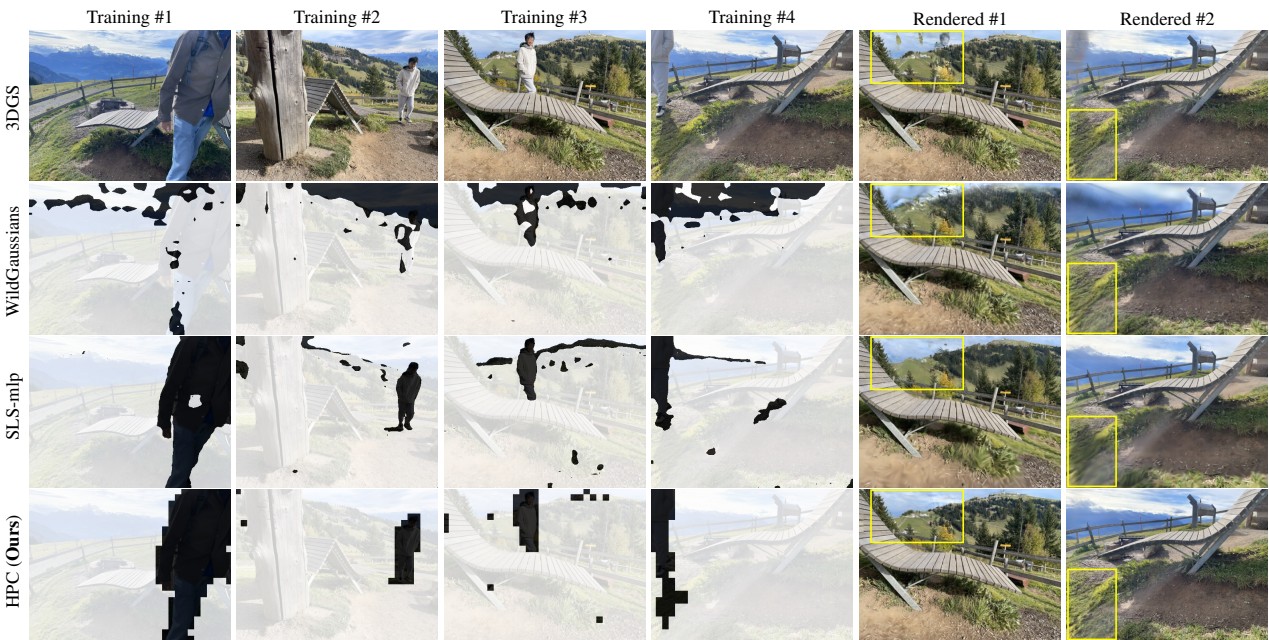

*Figure 1.* **Comparison between previous methods and our proposed Hybrid Patch-wise Classification (HPC).** When training 3DGS with static scenes disturbed by transient distractors, previous methods utilize image semantic priors but fail to separate transient distractors (*e.g.*, pedestrians and shadows) from static backgrounds (*e.g.*, mountains). By addressing their inherent semantic limitations, our method can generate binary masks that better represent transient distractors, thereby removing related artifacts in rendered results (yellow boxes).

## Abstract

3D Gaussian Splatting (3DGS) has demonstrated remarkable performance in novel view synthesis and 3D scene reconstruction, but its quality often degrades in real-world environments due to transient distractors, such as moving objects and varying shadows. Existing methods commonly introduce semantic priors from pre-trained vision models either to group pixels into coherent regions or to define perceptual error metrics. However, semantic grouping is often misaligned with the binary static/transient distinction, while perceptual features can be fragile under appearance perturbations introduced during 3DGS optimization. We propose 3DGS-HPC, a framework that addresses these issues by combining two complementary principles: a patch-wise classification strategy that leverages local spatial consistency for robust region-level decisions, and a hybrid classification metric that adaptively integrates photometric and perceptual cues for more reliable separation. Extensive experiments demonstrate the superiority and robustness of our method in mitigating distractors to improve 3DGS-based novel view synthesis. Our project page is https://cnhaox.github.io/3DGS-HPC/.

[1]Sun Yat-sen University, Guangzhou, China [2]Texas A&M University, College Station, USA [3]Cardiff University, Wales, UK [4]Centre for Perceptual and Interactive Intelligence, Hong Kong, China [5]Shenzhen Loop Area Institute, Shenzhen, China. Correspondence to: Guanbin Li <liguanbin@mail.sysu.edu.cn>.

*Proceedings of the 43rd International Conference on Machine Learning*, Seoul, South Korea. PMLR 306, 2026. Copyright 2026 by the author(s).

## 1. Introduction

Recent advances in 3D Gaussian Splatting (3DGS) (Huang et al., 2024a; Kerbl et al., 2023; Kheradmand et al., 2024; Yu

et al., 2024) have enabled high-quality novel view synthesis with real-time performance, establishing it as a strong alternative to Neural Radiance Fields (NeRF) (Mildenhall et al., 2020). However, similar to NeRF, 3DGS is built upon the assumption that all training images capture a completely static scene. In real-world settings, this assumption is frequently violated by *transient distractors* (*i.e.*, dynamic or temporally inconsistent elements such as pedestrians, vehicles, or shadows) that introduce artifacts and significantly degrade reconstruction quality. The diversity and unpredictability of these distractors in appearance, scale, and motion make them difficult to detect and remove reliably. As a result, transient distractors remain a key obstacle to deploying 3DGS for robust real-world scene reconstruction.

To mitigate the effects of distractors, most existing methods follow a simple yet effective paradigm: generating pixel-wise binary masks to label and filter out transient pixels in the training images, thereby preventing the model from being corrupted by distractors during optimization. This formulation reduces the task to estimating accurate binary masks, essentially a binary classification problem. Current research addresses this problem from two perspectives: *classification granularity* and *classification metric*. Early methods (Chen et al., 2022; Martin-Brualla et al., 2021; Sabour et al., 2023; Yang et al., 2023a) perform pixel-wise classification using photometric errors, but the resulting masks are often noisy and spatially inconsistent, as pixel-level predictions lack sufficient local context. Recent studies introduce semantic priors to improve both aspects. Some works (Chen et al., 2024; Kulhanek et al., 2024; Lee et al., 2023; Park et al., 2025; Ren et al., 2024; Sabour et al., 2025) enforce semantic coherence by unifying pixel-wise classification results into semantically consistent regions, thus enhancing spatial stability, while others (Kulhanek et al., 2024; Markin et al., 2024) adopt perceptual metrics that measure semantic similarity between training and rendered images. However, as shown in Fig. 1, the former use as an intra-image grouping prior relies on semantic regions derived from general vision models that are not necessarily aligned with the desired static/transient partitions, resulting in a *semantic mismatch* between predicted and ground-truth transient areas. The latter use remains valuable as a cross-image classification cue but can still suffer from *semantic fragility*, as feature representations are highly sensitive to minor appearance perturbations, leading to unstable responses and the misclassification of static regions as transient.

In this work, we address the challenges of semantic mismatch and fragility in existing methods by introducing a novel framework for distractor-free 3DGS, termed *Hybrid Patch-wise Classification (HPC)*. From the perspective of classification granularity, HPC avoids using external semantic priors for intra-image region grouping and instead leverages the assumption of *local spatial consistency*. By directly performing classification at the patch level, it captures richer local context than pixel-wise classification while maintaining higher robustness and efficiency than semantic region grouping. In terms of the classification metric, HPC keeps perceptual features as a useful cue but calibrates them with photometric evidence, yielding more reliable separation between static and transient regions. Extensive experiments demonstrate that HPC significantly improves the robustness and fidelity of 3DGS reconstructions in the presence of diverse distractors.

Our contributions are summarized as follows:

- We propose *Hybrid Patch-wise Classification (HPC)*, a novel framework for distractor-free 3D Gaussian Splatting (3DGS) that achieves robust reconstruction in real-world scenes containing transient distractors.

- To overcome the limitations of previous semantic-based methods, HPC introduces two complementary designs: a *patch-wise classification approach* that leverages local spatial consistency for efficient and context-aware classification, and a *hybrid classification metric* that adaptively fuses photometric and perceptual cues to enhance static/transient separation.

- We validate the effectiveness of HPC through extensive experiments, showing consistent improvements over state-of-the-art methods in both reconstruction quality and robustness to distractors.

## 2. Related Work

**NeRF & 3DGS in Non-static Scenes.** NeRF (Mildenhall et al., 2020) has emerged as a promising solution to achieve high-quality 3D scene reconstruction and novel view synthesis, while 3DGS (Kerbl et al., 2023) has been proposed as a compelling alternative due to its real-time rendering speed and comparable rendering quality. However, both approaches and many of their variants assume that the scene to be reconstructed is static and are therefore not applicable to real-world scenes containing non-static elements. In contrast to methods that explicitly reconstruct dynamic components (Duan et al., 2024; Huang et al., 2024b; Li et al., 2024; Luiten et al., 2024; Wu et al., 2024; Yang et al., 2023b; 2024), our work follows another research line that treats these non-static elements as transient distractors and removes them to enhance the reconstruction of static content. Early attempts (Liu et al., 2023; Rematas et al., 2022; Schischka et al., 2024; Sun et al., 2022; Tancik et al., 2022; Turki et al., 2022) used pre-trained semantic segmentation models (Chen et al., 2017a;b; 2018; Cheng et al., 2022) to detect transient distractors of known classes, but this prior information is intractable to obtain in practice. Subsequent research has focused on more generalizable approaches,

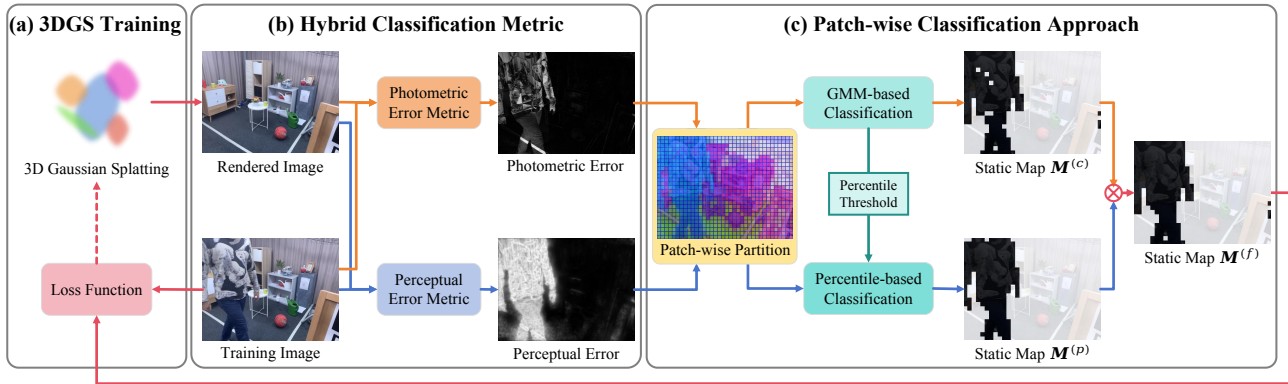

*Figure 2.* **Overview of Hybrid Patch-wise Classification (HPC).** Given training images disturbed by transient distractors, our method (a) optimizes 3DGS following standard practice, (b) classifies static and transient regions using a novel hybrid metric that combines the strengths of photometric and perceptual cues, and (c) performs classification at the patch level to exploit local spatial consistency, thereby enhancing robustness and efficiency without relying on external semantic segmentation or detection priors for region grouping.

which can be roughly grouped into two paradigms: uncertainty modeling (Chen et al., 2022; Dahmani et al., 2024; Kulhanek et al., 2024; Lee et al., 2023; Lin et al., 2025; Markin et al., 2024; Martin-Brualla et al., 2021; Park et al., 2025; Ren et al., 2024; Wang et al., 2024a;c; Yang et al., 2023a; Zhang et al., 2024), which relies on neural networks to predict pixel uncertainties of training images; and pixel classification (Bao et al., 2024; Chen et al., 2024; Sabour et al., 2023; 2025; Ungermann et al., 2024; Wang et al., 2024b; Xu et al., 2024), which directly classifies pixels as static or transient based on their reconstruction errors. We view the former as a soft form of the latter, as both rely on the Memorization Effect (Arazo et al., 2019; Arpit et al., 2017; Song et al., 2022; Wang et al., 2024b) that neural networks tend to learn contextually consistent samples faster than inconsistent samples. Consequently, pixels with larger reconstruction errors or higher uncertainties are regarded as transient and excluded from optimization during training.

**Semantics in Distractor-free NeRF & 3DGS.** Recent advances in vision foundation models (Caron et al., 2021; Kirillov et al., 2023; Oquab et al., 2023; Ravi et al., 2025; Tang et al., 2023) have inspired a new line of distractor-free NeRF and 3DGS methods that exploit semantic priors to better separate transient distractors from static backgrounds. These semantic priors are typically utilized in two ways:

- *Enhancing semantically coherent classification.* To encourage semantically similar pixels to produce coherent classification output, some methods (Fu et al., 2025; Kulhanek et al., 2024; Markin et al., 2024; Ren et al., 2024; Sabour et al., 2025) incorporate image semantics as an auxiliary input to the prediction network, while others (Bao et al., 2024; Chen et al., 2024; Markin et al., 2024; Park et al., 2025; Sabour et al., 2025) segment the image into semantically consistent regions and use them to refine classification results.

- *Serving as perceptual error metrics.* To address the

limitations of photometric error metrics (*e.g.*, RGB L1 error, SSIM error) in distinguishing distractors from static content with similar colors or textures, recent studies (Fu et al., 2025; Kulhanek et al., 2024; Markin et al., 2024) adopt perceptual metrics derived from feature embeddings to capture semantic-level differences.

Although these approaches achieve notable improvements, their reliance on external semantics introduces two challenges: *semantic mismatch* between general-purpose vision semantics and the specific static/transient distinction, and *semantic fragility* that small appearance perturbations may lead to unstable feature responses and misclassification. In our work, we address this gap by proposing a novel distractor-free 3DGS framework, which consists of i) a patch-wise classification approach that leverages local spatial consistency to improve robustness and efficiency without relying on external semantics for region grouping, and ii) a hybrid classification metric that combines perceptual and photometric error cues to correct semantic perturbability.

## 3. Preliminaries

Given multi-view images $\left\{\mathbf{I}_i \in \mathbb{R}^{H \times W \times 3}\right\}_{i=1}^{N_I}$, we have:

**3D Gaussian Splatting (3DGS).** 3DGS (Kerbl et al., 2023) represents the scene as a set of learnable 3D Gaussians $\mathcal{G} = \{\mathbf{g}_j\}_{j=1}^{N_G}$, each characterized by its position, covariance, color coefficients, and opacity. Given the specific camera parameters $\mathbf{P}_i$, 3DGS render the corresponding image $\hat{\mathbf{I}}_i$ in one forward pass instead of pixel batches:

$$\hat{\mathbf{I}}_i = F_{\text{render}}\left(\mathcal{G}, \mathbf{P}_i\right), \qquad (1)$$

where $F_{\text{render}}$ is the rendering function that performs 2D projection, tile splitting, depth-based sorting, and $\alpha$-blending. During training, $\mathcal{G}$ is optimized by minimizing the photometric error between the rendered image $\hat{\mathbf{I}}_i$ and the training

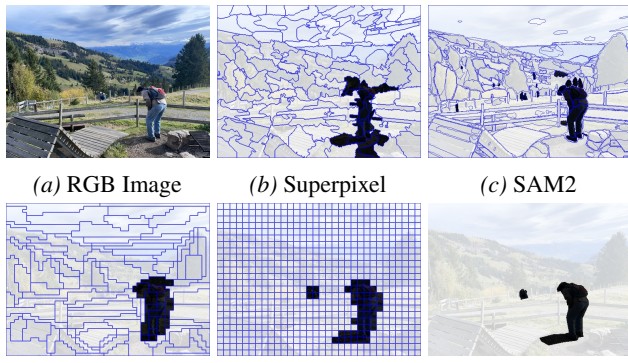

*(a)* RGB Image   *(b)* Superpixel   *(c)* SAM2

*(d)* Feature Clustering   *(e)* Patch-wise   *(f)* Reference Map

*Figure 3.* **Visualization of Different Partitioning Strategies.** When grouping pixels into multiple semantically consistent regions with blue borders, existing methods (b-d) fail to faithfully represent transient regions (*e.g.*, pedestrians and shadows) due to semantic mismatch between different tasks, while our patch-wise partition strategy (e) can accurately extract these regions.

image $\mathbf{I}_i$ using the following loss $\mathcal{L}$:

$$\mathcal{L}(\hat{\mathbf{I}}_i, \mathbf{I}_i) = (1 - \lambda)\mathcal{L}_1(\hat{\mathbf{I}}_i, \mathbf{I}_i) + \lambda\mathcal{L}_{\text{SSIM}}(\hat{\mathbf{I}}_i, \mathbf{I}_i), \quad (2)$$

where $\lambda = 0.2$; $\mathcal{L}_1$ and $\mathcal{L}_{\text{SSIM}}$ are L1 loss and SSIM loss.

**3DGS in Non-static Scenes.** An effective strategy to avoid artifacts caused by transient distractors is to mask out pixels related to distractors (*i.e.*, transient pixels) during training. Following existing work (Chen et al., 2024; Sabour et al., 2023; 2025), we introduce a binary map $\mathbf{M}_i \in \{0, 1\}^{H \times W}$ for each image $\mathbf{I}_i$, where 1 indicates static pixels and 0 denotes transient ones. We then apply $\mathbf{M}_i$ as masks to Eq. (2) to suppress distractor regions:

$$\mathcal{L}(\hat{\mathbf{I}}'_i, \mathbf{I}'_i) = (1 - \lambda)\,\mathcal{L}_1(\hat{\mathbf{I}}'_i, \mathbf{I}'_i) + \lambda\mathcal{L}_{\text{SSIM}}(\hat{\mathbf{I}}'_i, \mathbf{I}'_i), \quad (3)$$

where $\hat{\mathbf{I}}'_i = \mathbf{M}_i \odot \hat{\mathbf{I}}_i$ and $\mathbf{I}'_i = \mathbf{M}_i \odot \mathbf{I}_i$. Thus, the quality of reconstruction critically depends on the accuracy of the binary maps, which our method aims to improve.

## 4. Method

As Eq. (3) implies, the performance of 3DGS depends on the quality of binary maps, which are generated by classifying each pixel as static or transient. To this end, we propose a novel *Hybrid Patch-wise Classification (HPC)* framework (Fig. 2) that addresses the classification challenge from two complementary perspectives: the classification approach (Sec. 4.1) and the classification metric (Sec. 4.2) as follows.

### 4.1. Patch-wise Classification Approach

Recent methods have increasingly focused on incorporating semantic consistency to refine noisy pixel-wise classification results by grouping pixels into regions with coherent internal semantics. This is typically achieved using super-pixel algorithms (Van den Bergh et al., 2015), promptable

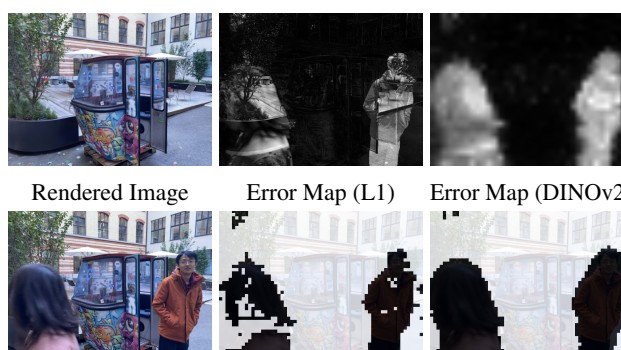

Rendered Image   Error Map (L1)   Error Map (DINOv2)

Trained Image   Static Map (L1)   Static Map(DINOv2)

*Figure 4.* **Visualization of Different Error Metrics.** Given training images with distractors and corresponding rendered images, traditional photometric metrics (L1) produce noisy static maps due to low-level appearance ambiguities (*e.g.*, similar colors). Instead, perceptual metrics based on vision foundation models (DINOv2) produce clean static maps with clear boundaries.

object segmentation (Kirillov et al., 2023; Ravi et al., 2025), or semantic feature clustering (Müllner, 2011; Tang et al., 2023). The motivation behind this trend stems from the observation that although static and transient pixels exhibit bimodal error distributions during training (Sabour et al., 2023; Wang et al., 2024b), accurate separation remains difficult in practice due to the limited context at the pixel level, leading to sensitivity to local perturbations and inconsistent predictions. Despite their success, these methods overlook the subtle differences in semantics between distractor-free 3DGS and other tasks (*e.g.*, semantic segmentation), thus producing suboptimal results.

**Semantic Mismatch in Region Grouping.** As illustrated by the binary map in Eq. (3), distractor-free 3DGS requires binary static/transient labels. However, vision foundation models provide semantics optimized for general visual understanding rather than this binary separation, and no existing method can reliably extract these specific semantics. As a result, existing approaches adopt workaround solutions by leveraging semantics defined for other tasks, under the assumption that static and transient regions can be approximated as a union of semantic regions for other tasks. For instance, as shown in Fig. 3, superpixel methods cluster pixels based on low-level appearance cues and are easily confused by color ambiguities (*e.g.*, black hairs *vs.* dark forests); SAM2 (Ravi et al., 2025) segments objects via prompt-based priors and tends to overlook subtle transient cues (*e.g.*, merging human shadows into the same ground region); and feature clustering groups pixels by similarity in the embedding space of vision foundation models, which emphasizes semantic similarity rather than temporal or photometric consistency, resulting in misclassifications similar to those of SAM2. Although each provides a form of semantic partitioning, none of them faithfully captures the binary semantic distinction required for distractor-free

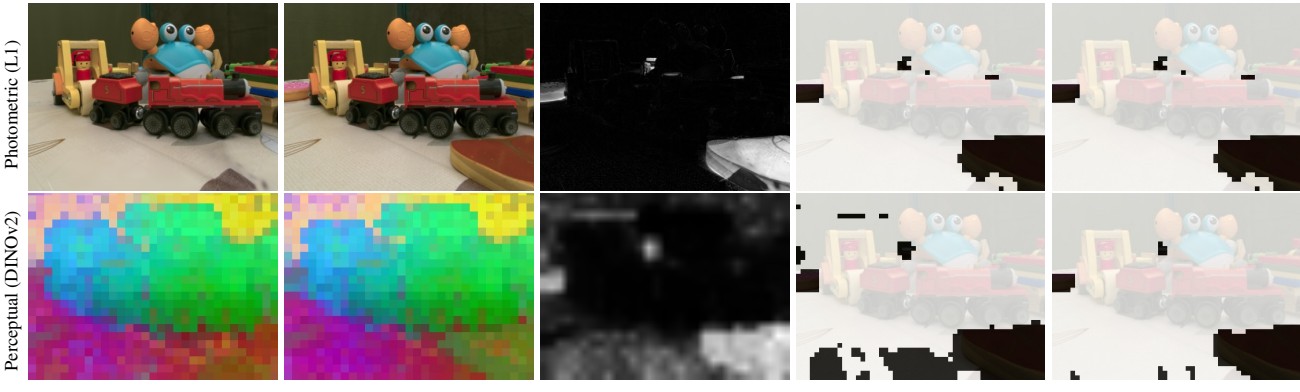

Rendered Image / Feature  Training Image / Feature  Error Map  Static Map (GMM)  Static Map (Percentile)

*Figure 5.* **Comparison of different classification approaches with different error metrics.** Bottom row: Although perceptual error metrics are effective in capturing semantic differences, they can produce anomalous results in textureless regions (*e.g.*, green wall and white tablecloth) due to visual perturbability (*e.g.*, blurring) between training and rendered images. Top row: In contrast, while inherently noisier, photometric error metrics demonstrate greater reliability in estimating the overall proportion of transient pixels within an image, thus guiding perception error metrics for more accurate classification.

3DGS, leading to mismatches and reduced performance.

**Patch-wise Partition Strategy.** To bridge this gap, we propose a simple yet effective alternative that avoids using external semantics for intra-image region grouping. Instead of relying on task-specific semantic grouping, we leverage the assumption of *local spatial consistency* (*i.e.*, neighboring pixels within a small region tend to share the same property) to overpartition each training image into regular, non-overlapping patches and directly perform classification at the patch level. Patch-wise classification provides richer contextual cues than pixel-wise classification and greater robustness to local perturbations, while significantly reducing computational cost by aggregating pixels into fewer units to be classified. Although this mechanical partition may produce boundaries that do not align precisely with the edges of the object, existing work (Markin et al., 2024) has demonstrated that reconstruction quality remains largely unaffected, provided that distractors are effectively identified.

Concretely, given a pixel-wise error map $\varepsilon_i \in \mathbb{R}^{H \times W}$ for image $\mathbf{I}_i$, we reshape it into a sequence of non-overlapping patches $\hat{\varepsilon}_i \in \mathbb{R}^{N_E \times P^2}$, where $(P, P)$ is the patch resolution and $N_E = HW/P^2$ is the number of patches. The mean error of each patch forms a compact patch-wise error sequence $\bar{\varepsilon}_i \in \mathbb{R}^{N_E}$ for classification.

**Classification Approaches (Percentile & GMM).** After aggregation, we can classify patches based on their total errors $\mathcal{E} \in \mathbb{R}^{N_I \cdot N_E}$ using two approaches:

- *Percentile-based Classification.* If the static proportion $\mathcal{T}$ of the scene is known, transient patches can be filtered using a percentile threshold:

$$\mathbf{M}_i(j) = \mathbb{I}\left[\bar{\varepsilon}_i(j) \leq \text{percentile}\left(\mathcal{E}, \mathcal{T}\right)\right], \quad (4)$$

where $\mathbf{M}_i(j)$ is the static mask of the $j$-th patch of

the $i$-th image, and $\mathbb{I}[\cdot]$ is the indicator function that returns 1 if the predicate is true and 0 otherwise.

- *GMM-based Classification.* In practice, the exact static proportion varies with different scenes and is hard to obtain in advance. To achieve accurate distinction in various settings, we follow prior works (Li et al., 2020; Wang et al., 2024b) to fit the error distribution via a two-component one-dimensional Gaussian Mixture Model (GMM). The probability density function $P\left(\bar{\varepsilon}_i(j)\right)$ is:

$$P\left(\bar{\varepsilon}_i(j)\right) = \sum_{k=1}^{2} \beta_k \cdot \mathcal{N}\left(\bar{\varepsilon}_i(j) \mid \mu_k, \sigma_k^2\right), \quad (5)$$

where $\beta_k$ represents the mixing component of the $k$-th gaussian distribution $\mathcal{N}\left(\cdot \mid \mu_k, \sigma_k^2\right)$ with the mean $\mu_k$ and the variance $\sigma_k^2$. Let $\mu_1 \leq \mu_2$, we can view the first distribution as static, and determine whether the $j$-th patch of the $i$-th image is static by the probability that its error value $\bar{\varepsilon}_i(j)$ belongs to the static distribution:

$$\mathbf{M}_i(j) = \mathbb{I}\left[\frac{\beta_1 \cdot \mathcal{N}\left(\bar{\varepsilon}_i(j) \mid \mu_1, \sigma_1^2\right)}{\sum_{k=1}^{2} \beta_k \cdot \mathcal{N}\left(\bar{\varepsilon}_i(j) \mid \mu_k, \sigma_k^2\right)} \geq 0.5\right]. \quad (6)$$

### 4.2. Hybrid Classification Metrics

While patch-wise classification improves spatial granularity and robustness, the effectiveness of distractor removal also depends on the reliability of the underlying classification metric. Recent state-of-the-art methods replace traditional photometric error metrics with perceptual error metrics (*e.g.*, DINOv2 (Oquab et al., 2023)) to capture object-level inconsistencies between reference and rendered images. As shown in Fig. 4, these perceptual metrics offer clear advantages: they enable image comparison at the *semantic* level while exhibiting robustness to ambiguities at the *appearance* level (*e.g.*, objects with similar colors). Rather

| Method | Statue | | | Android | | | Yoda | | | Crab (2) | | | Avg. | | |
|---|---|---|---|---|---|---|---|---|---|---|---|---|---|---|---|
| | PSNR↑ | SSIM↑ | LPIPS↓ | PSNR↑ | SSIM↑ | LPIPS↓ | PSNR↑ | SSIM↑ | LPIPS↓ | PSNR↑ | SSIM↑ | LPIPS↓ | PSNR↑ | SSIM↑ | LPIPS↓ |
| RobustNeRF (Sabour et al., 2023) | 21.56 | .841 | .129 | 24.38 | .828 | .088 | 33.72 | .940 | .079 | 32.77 | .934 | .077 | 28.11 | .886 | .093 |
| NeRF-HuGS (Chen et al., 2024) | 22.14 | .850 | .120 | 24.40 | .834 | .088 | 33.64 | .945 | .057 | 33.83 | .939 | .061 | 28.50 | .892 | .081 |
| NeRF On-the-go (Ren et al., 2024) | 22.34 | .849 | .109 | 24.42 | .823 | .084 | 30.64 | .933 | .068 | 29.65 | .917 | .079 | 26.76 | .881 | .085 |
| 3DGS (Kerbl et al., 2023) | 21.54 | .848 | .135 | 23.62 | .814 | .097 | 30.34 | .945 | .086 | 30.35 | .944 | .084 | 26.46 | .888 | .101 |
| MemE-3DGS (Wang et al., 2024b) | 22.65 | .851 | .096 | 24.14 | .810 | .079 | 31.41 | .928 | .115 | 32.05 | .940 | .093 | 27.56 | .882 | .096 |
| WildGaussians (Kulhanek et al., 2024) | 23.02 | .861 | .127 | 25.15 | .837 | .092 | 31.91 | .944 | .111 | 31.42 | .936 | .114 | 27.88 | .894 | .111 |
| SLS-mlp (Sabour et al., 2025) | 22.81 | .841 | .119 | 25.04 | .834 | .079 | 35.52 | .957 | .071 | 34.66 | .953 | .076 | 29.51 | .896 | .086 |
| HybridGS (Lin et al., 2025) | 22.87 | .862 | .104 | 25.14 | .848 | .068 | 35.19 | .958 | .073 | 34.94 | .955 | .074 | 29.53 | .906 | .080 |
| T-3DGS (Markin et al., 2024) | 22.68 | .855 | .107 | 24.67 | .822 | .073 | 33.02 | .942 | .095 | 33.11 | .943 | .092 | 28.37 | .890 | .092 |
| RobustSplat (Fu et al., 2025) | 22.80 | .865 | .110 | 24.62 | .831 | .125 | 35.14 | .944 | .151 | 34.88 | .940 | .154 | 29.36 | .895 | .135 |
| Ours (VGG) | 23.04 | .878 | .096 | 25.12 | .853 | .064 | 35.78 | .962 | .068 | 35.43 | .960 | .066 | 29.84 | .913 | .074 |
| Ours (ResNet) | 23.09 | .879 | .095 | 25.01 | .851 | .065 | 35.88 | .962 | .068 | 35.49 | .960 | .066 | 29.87 | .913 | .073 |
| Ours (DINOv2) | 23.38 | .880 | .094 | 25.02 | .851 | .065 | 35.79 | .962 | .068 | 35.07 | .957 | .071 | 29.81 | .913 | .074 |

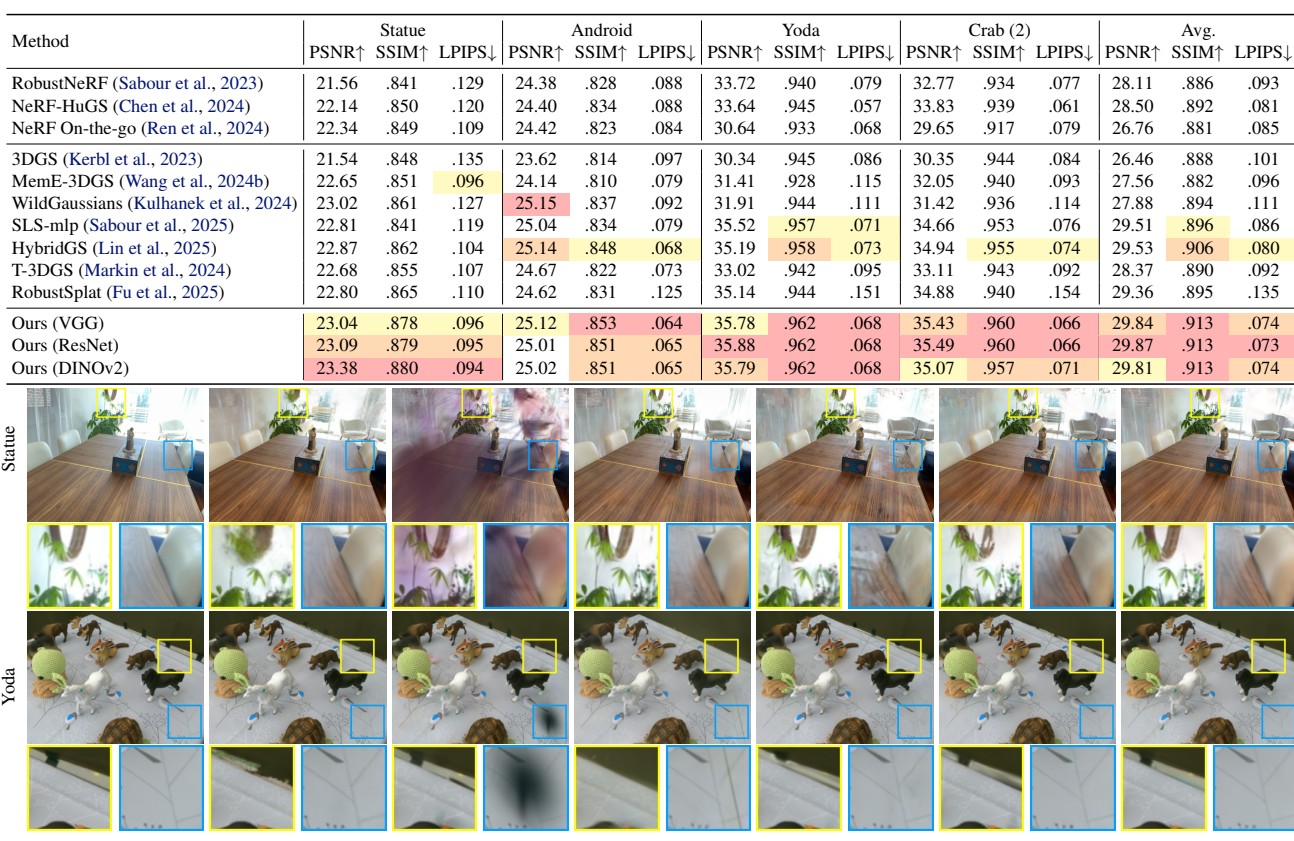

Ground Truth    NeRF-HuGS    3DGS    WildGaussians    SLS-mlp    HybridGS    Ours (DINOv2)

*Figure 6.* **Quantitative and Qualitative Comparisons on the RobustNeRF Dataset (Sabour et al., 2023).** The first , second , and third best values are highlighted. Quantitatively, our method not only improves the performance of native 3DGS, but also outperforms previous methods on most quality metrics. Qualitatively, our method can better preserve static details while ignoring distractors.

than contributing to this line of work by designing a better perceptual metric, our study takes a step back to critically examine the paradigm itself. Specifically, we identify an inherent limitation shared by existing perceptual metrics: *semantic fragility in specific image regions*. To address this, we propose a novel hybrid classification metric that synergistically combines perceptual and photometric cues, leading to more robust and accurate static map construction.

**Semantic Fragility of Perceptual Metrics.** Despite their effectiveness, perceptual metrics can still yield anomalous results due to inconsistent semantic predictions of vision foundation models for minor appearance perturbations (*e.g.*, blurring or color jitter). Since feature extractors in these models are optimized for object-centric understanding rather than appearance consistency, their outputs are tuned to salient object-related features, leading to unstable activations in textureless regions. This issue is especially pronounced in low-frequency regions during 3DGS optimization, where appearance perturbations between training and predicted images can dominate and result in abnormally high perceptual errors, causing static areas to be incorrectly classified as transient (Fig. 5). Given that low-texture regions constitute

a significant portion of natural images (Buccigrossi & Simoncelli, 1999; Cho et al., 2010; Li & Orchard, 2001; Lyu, 2005), perceptual metrics are therefore inherently prone to erroneously filter out a greater number of static regions compared to their photometric counterparts.

**Hybrid Metric Formulation.** We therefore propose a hybrid classification metric that combines perceptual and photometric cues. In this framework, the photometric error (processed via GMM-based classification) provides an estimated static pixel proportion, which guides the percentile thresholding used in perceptual error-based classification. By jointly measuring image differences in both color and semantic spaces, this hybrid approach not only improves classification accuracy but also enables adaptive control over the static proportion within a reasonable range.

Specifically, we first obtain the photometric error map $\varepsilon_i^{(c)}$ and the perceptual error map $\varepsilon_i^{(p)}$ between the rendered image $\hat{\mathbf{I}}_i$ and the training image $\mathbf{I}_i$ by using the RGB L1 loss and the feature cosine similarity loss, respectively:

$$\varepsilon_i^{(c)} = \|\hat{\mathbf{I}}_i - \mathbf{I}_i\|_1, \quad \varepsilon_i^{(p)} = 1 - \frac{\hat{\mathbf{f}}_i \cdot \mathbf{f}_i}{\|\hat{\mathbf{f}}_i\|_2 \cdot \|\mathbf{f}_i\|_2}, \quad (7)$$

| Method | Low Occlusion | | | | | | Medium Occlusion | | | | | | High Occlusion | | | | | | Avg. | | |
| | Mountain | | | Fountain | | | Corner | | | Patio | | | Spot | | | Patio-High | | | | | |
| | PSNR↑ | SSIM↑ | LPIPS↓ | PSNR↑ | SSIM↑ | LPIPS↓ | PSNR↑ | SSIM↑ | LPIPS↓ | PSNR↑ | SSIM↑ | LPIPS↓ | PSNR↑ | SSIM↑ | LPIPS↓ | PSNR↑ | SSIM↑ | LPIPS↓ | PSNR↑ | SSIM↑ | LPIPS↓ |
|---|---|---|---|---|---|---|---|---|---|---|---|---|---|---|---|---|---|---|---|---|---|
| RobustNeRF (Sabour et al., 2023) | 21.37 | .732 | .151 | 20.97 | .717 | .191 | 24.30 | .863 | .099 | 21.33 | .781 | .127 | 21.83 | .686 | .372 | 19.78 | .632 | .317 | 21.60 | .735 | .210 |
| NeRF-HuGS (Chen et al., 2024) | 21.65 | .741 | .148 | 21.44 | .732 | .174 | 26.52 | .879 | .087 | 22.06 | .812 | .108 | 23.06 | .823 | .157 | 22.45 | .776 | .173 | 22.86 | .794 | .141 |
| NeRF On-the-go (Ren et al., 2024) | 21.27 | .719 | .148 | 16.11 | .482 | .429 | 25.96 | .865 | .086 | 21.47 | .793 | .117 | 24.06 | .865 | .088 | 22.25 | .790 | .158 | 21.85 | .752 | .171 |
| 3DGS (Kerbl et al., 2023) | 20.79 | .749 | .153 | 21.27 | .773 | .157 | 22.77 | .832 | .154 | 17.66 | .715 | .204 | 18.60 | .737 | .389 | 18.09 | .701 | .296 | 19.86 | .751 | .225 |
| MemE-3DGS (Wang et al., 2024b) | 20.45 | .699 | .173 | 21.70 | .744 | .152 | 25.26 | .867 | .081 | 16.81 | .728 | .183 | 23.68 | .880 | .121 | 20.68 | .764 | .187 | 21.43 | .780 | .149 |
| WildGaussians (Kulhanek et al., 2024) | 20.57 | .706 | .224 | 22.33 | .757 | .180 | 26.13 | .881 | .102 | 21.82 | .823 | .126 | 24.96 | .867 | .121 | 23.29 | .814 | .157 | 23.18 | .808 | .152 |
| SLS-mlp (Sabour et al., 2025) | 21.99 | .726 | .189 | 22.48 | .760 | .168 | 26.33 | .889 | .090 | 22.50 | .828 | .108 | 24.34 | .855 | .132 | 23.15 | .805 | .152 | 23.47 | .810 | .140 |
| HybridGS (Lin et al., 2025) | 21.26 | .618 | .348 | 20.68 | .631 | .251 | 24.84 | .817 | .119 | 21.90 | .784 | .149 | 23.81 | .712 | .191 | 21.59 | .689 | .216 | 22.35 | .708 | .212 |
| T-3DGS (Markin et al., 2024) | 20.50 | .689 | .195 | 22.45 | .770 | .137 | 25.50 | .870 | .081 | 21.99 | .842 | .094 | 24.83 | .887 | .116 | 22.43 | .806 | .147 | 22.95 | .811 | .128 |
| RobustSplat (Fu et al., 2025) | 21.15 | .737 | .201 | 21.01 | .701 | .199 | 26.42 | .897 | .104 | 21.63 | .827 | .139 | 26.21 | .907 | .102 | 22.87 | .837 | .146 | 23.22 | .818 | .149 |
| Ours (VGG) | 21.64 | .778 | .134 | 23.31 | .812 | .121 | 25.18 | .898 | .082 | 22.69 | .846 | .096 | 24.99 | .902 | .091 | 23.54 | .851 | .120 | 23.56 | .848 | .107 |
| Ours (ResNet) | 22.40 | .786 | .128 | 23.58 | .815 | .117 | 25.44 | .899 | .079 | 22.64 | .846 | .094 | 25.75 | .904 | .090 | 23.17 | .851 | .120 | 23.83 | .850 | .105 |
| Ours (DINOv2) | 22.20 | .784 | .129 | 23.40 | .813 | .117 | 25.72 | .900 | .077 | 22.67 | .845 | .095 | 25.83 | .899 | .096 | 23.36 | .850 | .122 | 23.86 | .849 | .106 |

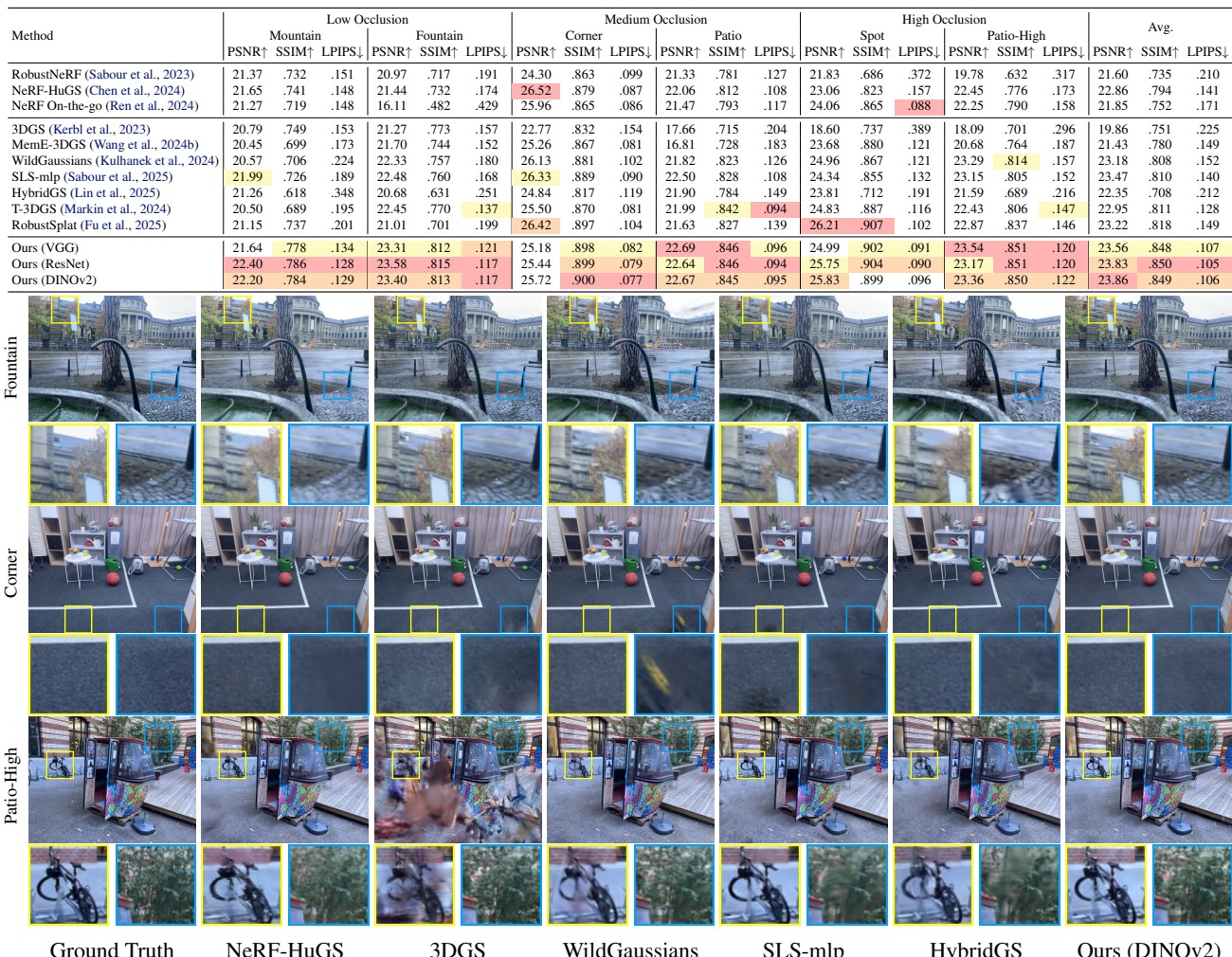

Ground Truth    NeRF-HuGS    3DGS    WildGaussians    SLS-mlp    HybridGS    Ours (DINOv2)

*Figure 7.* **Quantitative and Qualitative Comparisons on the On-the-go Dataset (Ren et al., 2024).** The first , second , and third best values are highlighted. Our method consistently achieves top-three quantitative results in scenes with varying distractor ratios. In terms of visual quality, our method can better reconstruct static content, while previous methods suffer from missing or distorted details.

where $\hat{\mathbf{f}}_i$ and $\mathbf{f}_i$ are feature maps extracted from vision foundation models. We follow the process in Sec. 4.1 to reorganize photometric error maps into the patch-wise error sequence, and then classify it through Eq. (6) to obtain the static map $\mathbf{M}_i^{(c)}$ and the estimated static proportion $\mathcal{T}^{(c)}$:

$$\mathcal{T}^{(c)} = \frac{\sum_{i=1}^{N_I} \sum_{j=1}^{N_E} \mathbf{M}_i^{(c)}(j)}{N_I \cdot N_E}. \tag{8}$$

Similarly, we reorganize the perceptual error maps and obtain their static map $\mathbf{M}_i^{(p)}$ through Eq. (4) with $\mathcal{T}^{(c)}$. The final static map $\mathbf{M}_i^{(f)}$ can be defined as:

$$\mathbf{M}_i^{(f)} = \mathbf{M}_i^{(c)} \cap \mathbf{M}_i^{(p)}. \tag{9}$$

In the end, the static map $\mathbf{M}_i^{(f)}$ is applied to Eq. (3) to filter out transient distractors during training.

# 5. Experiments

## 5.1. Experimental Setups

**Datasets.** We use two public datasets in our experiments:

- *RobustNeRF Dataset (Sabour et al., 2023).* This real-world dataset includes 4 controlled indoor scenes, each with 1-150 common objects placed as distractors. Following the baseline settings, we report results on the latest version (Sabour et al., 2025) and downsample images by 8x for training and evaluation.

- *On-the-go Dataset (Ren et al., 2024).* This real-world dataset contains multiple casually captured indoor and outdoor scenes, each with varying ratios of distractors (from 5% to over 30%). Following the baseline settings, we report results on the 6 selected scenes representing different occlusion rates, and downsample images by 8x (except images of *Patio* downsampled by 4x).

| | Partition | RobustNeRF (Avg.) | | | On-the-go (Avg.) | | |
|---|---|---|---|---|---|---|---|
| | | PSNR↑ | SSIM↑ | LPIPS↓ | PSNR↑ | SSIM↑ | LPIPS↓ |
| (a) | N/A | 29.67 | .911 | .077 | 22.95 | .834 | .127 |
| (b) | Superpixel (Van den Bergh et al., 2015) | 29.74 | .912 | .075 | 23.29 | .846 | .109 |
| (c) | SAM2 (Ravi et al., 2025) | 29.45 | .913 | .074 | 23.19 | .844 | .112 |
| (d) | Feat. Clustering (Tang et al., 2023) | 29.70 | .912 | .075 | 23.71 | .848 | .107 |
| (e) | Patch-wise (Ours) | 29.81 | .913 | .074 | 23.86 | .849 | .106 |

| | Photometric | Perceptual | Avg. Static% | RobustNeRF (Avg.) | | | On-the-go (Avg.) | | |
|---|---|---|---|---|---|---|---|---|---|
| | | | | PSNR↑ | SSIM↑ | LPIPS↓ | PSNR↑ | SSIM↑ | LPIPS↓ |
| (a) | GMM | - | 85.31 | 29.82 | .912 | .074 | 22.83 | .835 | .123 |
| (b) | - | GMM | 73.22 | 29.13 | .905 | .085 | 23.40 | .834 | .129 |
| (c) | - | Percentile | 86.26 | 29.55 | .912 | .074 | 23.31 | .845 | .116 |
| (d) | GMM | GMM | 72.62 | 29.48 | .909 | .083 | 23.06 | .825 | .135 |
| (e) | GMM | Percentile | 82.75 | 29.81 | .913 | .074 | 23.86 | .849 | .106 |

| Image | (a) | (b) | (c) | (d) | (e) |

*Figure 8.* **Comparisons of Partitioning Strategies.** Pixel-wise method (a) produces noisy static maps, while other methods (b-d) confuse the desktop with transient shadows. Instead, our method (e) accurately distinguishes between transient and static elements.

| Image | (a) | (b) | (c) | (d) | (e) |

*Figure 9.* **Comparisons of Classification Metrics.** (a) fails to distinguish distractors (*e.g.*, clothing) that closely resemble static backgrounds in color, while (b) tends to incorrectly remove certain static regions (*e.g.*, wall surfaces). A naive combination (d) does not yield meaningful improvements. In contrast, our method (e) effectively separates the transient region from the static one.

**Implementation Details.** Please see the appendix for more details. In short, we develop our method based on gsplat (Ye et al., 2025), and follow the same model architecture and training strategies as native 3DGS (Kerbl et al., 2023), except that a modified loss Eq. (3) is applied to filter out distractors. We warm up the model without applying static maps in the first 500 iterations and then update static maps every 100 iterations. For patch-wise classification approaches, we set the patch size to 16 by default. For perceptual error metrics, we separately apply three vision foundation models with different architectures: VGG16 (Simonyan & Zisserman, 2014), ResNet18 (He et al., 2016), and DINOv2 ViT-S/14 (Oquab et al., 2023). All comparisons use this default configuration without per-scene manual tuning.

### 5.2. Evaluation on Novel View Synthesis

**Baselines.** In addition to native 3DGS, we compare our method to three state-of-the-art NeRF-based methods and six recent 3DGS-based methods: RobustNeRF (Sabour et al., 2023), NeRF-HuGS (Chen et al., 2024), NeRF On-the-go (Ren et al., 2024), MemE-3DGS (Wang et al., 2024b), WildGaussians (Kulhanek et al., 2024), SLS-mlp (Sabour et al., 2025), HybridGS (Lin et al., 2025), T-3DGS (Markin et al., 2024), and RobustSplat (Fu et al., 2025). Among them, NeRF-HuGS, NeRF On-the-go, WildGaussians, SLS-mlp, T-3DGS, and RobustSplat utilize semantic priors (Kirillov et al., 2023; Oquab et al., 2023; Rombach et al., 2022) to enhance the results' semantic consistency, while WildGaussians, T-3DGS, and RobustSplat use DINOv2 cosine error as a perceptual metric. All methods are trained on images disturbed by distractors and evaluated on clean images. We report image synthesis quality based on PSNR, SSIM (Wang et al., 2004) and LPIPS (Zhang et al., 2018).

**Comparisons on the RobustNeRF Dataset (Fig. 6).** Compared to native 3DGS, applying our method leads to substantial PSNR improvements of 1.39 to 5.54 dB, indicating that our method can generate high-quality static maps to prevent

native 3DGS from being affected by transient distractors. Compared to other baselines, our method achieves almost all the best quantitative results and maintains a good balance between ignoring transient distractors (*e.g.*, pink and dark artifacts) and preserving static details (*e.g.*, the mirror and wall structure). Furthermore, our method does not show significant performance differences when using different vision foundation models, demonstrating the adaptability and generalizability of our method.

**Comparisons on the On-the-go Dataset (Fig. 7).** The results and conclusions are similar to those on the RobustNeRF dataset. Specifically, even with semantic priors, WildGaussians and SLS-mlp still fail to eliminate distractors (*e.g.*, temporal shadows) while incorrectly losing static details (*e.g.*, ground textures and the bicycle frame). Instead, our method achieves better visual quality by addressing the inherent mismatches and biases in external semantics to better separate transient distractors and static background.

### 5.3. Ablation Study

Based on our method with DINOv2, we test key components and hyperparameters to study their effects. More experiments and analyzes can be found in the appendix.

**Partitioning Strategies.** As shown in Fig. 8, method (a) without any partitioning strategy performs the worst. Methods (b-d) rely on low-level appearance cues or deep semantic priors to organize pixels, achieving suboptimal results. In contrast, our method (e) performs the best, indicating that the simple patch-wise strategy is sufficient for this task.

**Classification Metrics.** As shown in Fig. 9, using perceptual metrics alone (b) performs worse than using color metrics alone (a) in some cases, and their difference in estimated static proportions is aligned with the analysis of Sec. 4.2. Method (d), which simply combines them, cannot achieve better performance. By using the static proportion

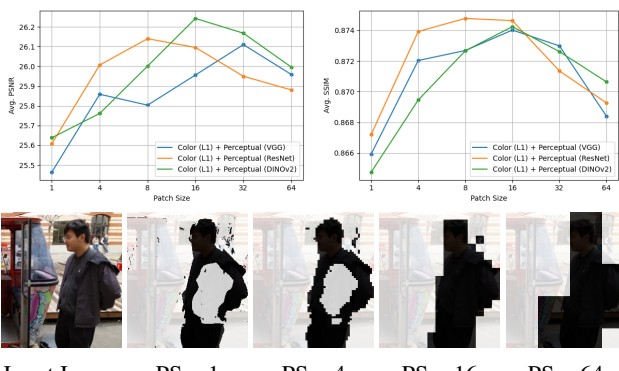

Input Image    PS = 1    PS = 4    PS = 16    PS = 64

*Figure 10.* **Performance of Different Patch Sizes (PS).** Patch sizes that are extremely small ($< 8$) or large ($> 16$) will lead to suboptimal quantitative results, and the resulting mask will fail to accurately label and filter out transient distractors (*e.g.*, people).

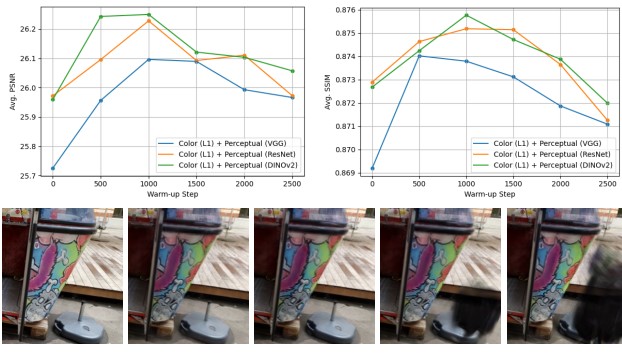

GT    WS = 0    WS = 500    WS = 1,500    WS = 2,500

*Figure 11.* **Performance of Different Warm-up Steps (WS).** When the number of warm-up steps exceeds $1,000$, the model is at risk of "memorizing" transient distractors, leading to artifacts in the rendered results.

estimated by color metrics to guide perceptual metrics for classification, our method (e) achieves the best results.

**Patch Size Sensitivity.** As shown in Fig. 10, the average PSNRs and SSIMs of our method using different perceptual error metrics first increase and then decrease as the patch size gradually increases on both the RobustNeRF and On-the-go datasets. This indicates that the best patch size is achieved by striking a balance between maintaining local spatial consistency and flexibly representing transient distractors. Nevertheless, our method consistently achieves better performance than pixel-wise classification (*i.e.*, Patch Size $= 1$) even without carefully choosing the patch size, consistent with the results above.

**Warm-up Step Sensitivity.** The warm-up phase aims to prevent static details from being excessively removed during early training, as the model will underfit the scene and have large errors when initialized. As shown in Fig. 11, the use of too few warm-up steps often degrades performance as it unnecessarily removes static details, while the use of too many steps causes 3DGS to overfit to transient distractors. We therefore set the default warm-up step to 500.

## 6. Conclusion

In conclusion, we presented Hybrid Patch-wise Classification (HPC), a novel framework for enabling distractor-free 3DGS. By combining a patch-wise classification strategy with a hybrid error metric that fuses perceptual and color cues, our method addresses two limitations in prior semantic-prior usage: mismatched semantic region grouping and uncalibrated perceptual classification. Extensive experiments demonstrate that HPC significantly improves reconstruction quality and robustness in the presence of transient distractors, offering a practical and generalizable solution for real-world 3DGS applications. Please see the appendix for **limitations and future work**.

## Acknowledgements

This work is supported in part by the National Key R&D Program of China (Nos.2024YFB3908503 and 2024YFB3908500), in part by the National Natural Science Foundation of China (No. 62322608), and in part by the Shenzhen Loop Area Institute under grant FPF10120260001.

## Impact Statement

This paper advances robust 3D reconstruction and novel view synthesis by improving 3D Gaussian Splatting in the presence of transient distractors (*e.g.*, moving objects and varying shadows). By producing more reliable static/transient masks, our approach can improve reconstruction quality from in-the-wild captures, which may benefit applications such as AR/VR content creation, robotics, and mapping in dynamic real-world environments.

Potential negative impacts are similar to those of 3D reconstruction systems more broadly: improved robustness may lower the barrier to reconstructing real spaces in privacy-sensitive contexts, and transient-object suppression could be misused to create misleading visualizations if not disclosed. We encourage deployment with appropriate consent and compliance with privacy regulations, and clear communication when transient content has been filtered.

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

# Appendix

*Table 1.* **Settings for Our Perceptual Error Metrics.**

| Vision Model | Extracted Feature Layers |
|---|---|
| VGG-16 (Simonyan & Zisserman, 2014) | {conv1_2, conv2_2, conv3_3, conv4_3, conv5_3} |
| ResNet-18 (He et al., 2016) | {layer1, layer2, layer3, layer4} |
| DINOv2 ViT-S/14 (Oquab et al., 2023) | {layer2, layer5, layer8, layer11} |

*Table 2.* **Performance of Our Patch-wise Classification *vs.* Its Refinement Variant.** Our method can significantly reduce training time and have slightly better performance in visual metrics.

| | RobustNeRF (Avg.) | | | | On-the-go (Avg.) | | | |
|---|---|---|---|---|---|---|---|---|
| | PSNR↑ | SSIM↑ | LPIPS↓ | Train hrs.↓ | PSNR↑ | SSIM↑ | LPIPS↓ | Train hrs.↓ |
| (a) Refinement Variant | 29.74 | .912 | **.074** | 2.53 | 23.69 | .848 | .111 | 2.09 |
| (b) Ours | **29.81** | **.913** | **.074** | **0.22** | **23.86** | **.849** | **.106** | **0.25** |

## A. Implementation & Experimental Details

**3DGS.** As mentioned in Sec. 5.1 of the main paper, we develop our model based on the open-source codebase gsplat (Ye et al., 2025), which reproduces native 3DGS (Kerbl et al., 2023) and aligns its performance. We follow the same training settings and adaptive density control strategy as the native model, except that we do not remove large-scale Gaussians during periodic pruning, which would otherwise result in some scene backgrounds represented by large-scale Gaussians having degraded geometry and appearance. All experiments of our method are conducted on an NVIDIA RTX 4090 GPU.

**Update of Static Maps.** Similar to prior works (Kulhanek et al., 2024; Markin et al., 2024), we do not apply or update static maps in the first 500 steps to warm up the 3DGS optimization and ensure the initialization of the scene reconstruction. We then apply static maps in the remaining steps and update them every 100 steps by rendering all training images with the current 3DGS. Since 3DGS resets the opacity every 3,000 steps during training, we only apply static maps but do not update them within 200 steps after each opacity reset.

**Perceptual Error Metrics.** Inspired by LPIPS (Zhang et al., 2018), we implement our perceptual metrics by extracting multi-layer features from vision foundation models. Since feature maps from different layers have varying spatial resolutions, we first compute cosine error maps at each layer and then upsample them to the input resolution using bilinear interpolation. The final error map is obtained by averaging these aligned maps. The vision models and their specific layers used for feature extraction are summarized in Tab. 1.

**Partitioning Strategies.** We implement different partitioning strategies for comparison:

- *Superpixel.* Following ForestSplats (Park et al., 2025), we leverage the superpixel algorithm SEEDS (Van den Bergh et al., 2015) from the OpenCV library and set

*Table 3.* **Quantitative Efficiency Comparisons between Different Methods.** Our method outperforms most other methods in training time and memory usage.

| | Avg. Train Time (min)↓ | Avg. GPU Mem (GB)↓ |
|---|---|---|
| SLS-mlp (Sabour et al., 2025) | 11.91 | 3.60 |
| WildGaussians (Kulhanek et al., 2024) | 24.10 | 16.02 |
| T-3DGS (Markin et al., 2024) | 26.75 | 4.36 |
| Ours (VGG) | 16.35 | 2.09 |
| Ours (ResNet) | 13.29 | 1.98 |
| Ours (DINOv2) | 17.52 | 2.05 |

*Table 4.* **Quantitative Comparisons between Different Types of GT Masks.** Inaccurate object boundaries in masks can still yield comparable results, whereas region misclassification cannot.

| | GT Mask Type | Avg. PSNR↑ | Avg. SSIM↑ | Avg. LPIPS↓ |
|---|---|---|---|---|
| (1) | Unchanged | 29.66 | .906 | .074 |
| (2) | Dilate 5 pixels | 30.01 | .908 | .074 |
| (3) | Convert to $16 \times 16$ patches | 29.93 | .906 | .077 |
| (4) | Misclassify 10% of (3) | 29.35 | .901 | .081 |

the superpixel number of each image to 100 by default.

- *SAM2.* The Segment Anything Model 2 (SAM2) (Ravi et al., 2025) is a foundation model extended from SAM (Kirillov et al., 2023) for promptable visual segmentation in images and videos. Following NeRF-HuGS (Chen et al., 2024), we take a regular point grid with a resolution of $64 \times 64$ as input prompt to generate instance masks for each training image.

- *Feature Clustering.* Following SpotLessSplats (Sabour et al., 2025), we execute agglomerative clustering (Müllner, 2011) on the feature map from the 2nd upsampling layer of Stable Diffusion v2.1 (Rombach et al., 2022; Tang et al., 2023), and set the cluster number of each image to 100 by default.

## B. Additional Experiments

### B.1. Patch-wise Classification Implementation

In Sec. 4.1 of the main paper, we propose to classify patches directly based on their average errors rather than using them to refine results after per-pixel classification. As shown in Tab. 2, when the patch size is set to 16, our proposed method significantly reduces training time due to the reduction in the number of classification samples, and performs slightly better in visual metrics. We therefore uniformly use our proposed patch-wise classification implementation in other experiments for efficiency.

### B.2. Efficiency Comparisons between Different Methods

As shown in Tab. 3, we additionally provide quantitative efficiency comparisons of different methods on all the scenes used in the main paper. Specifically, SLS-mlp (Sabour et al.,

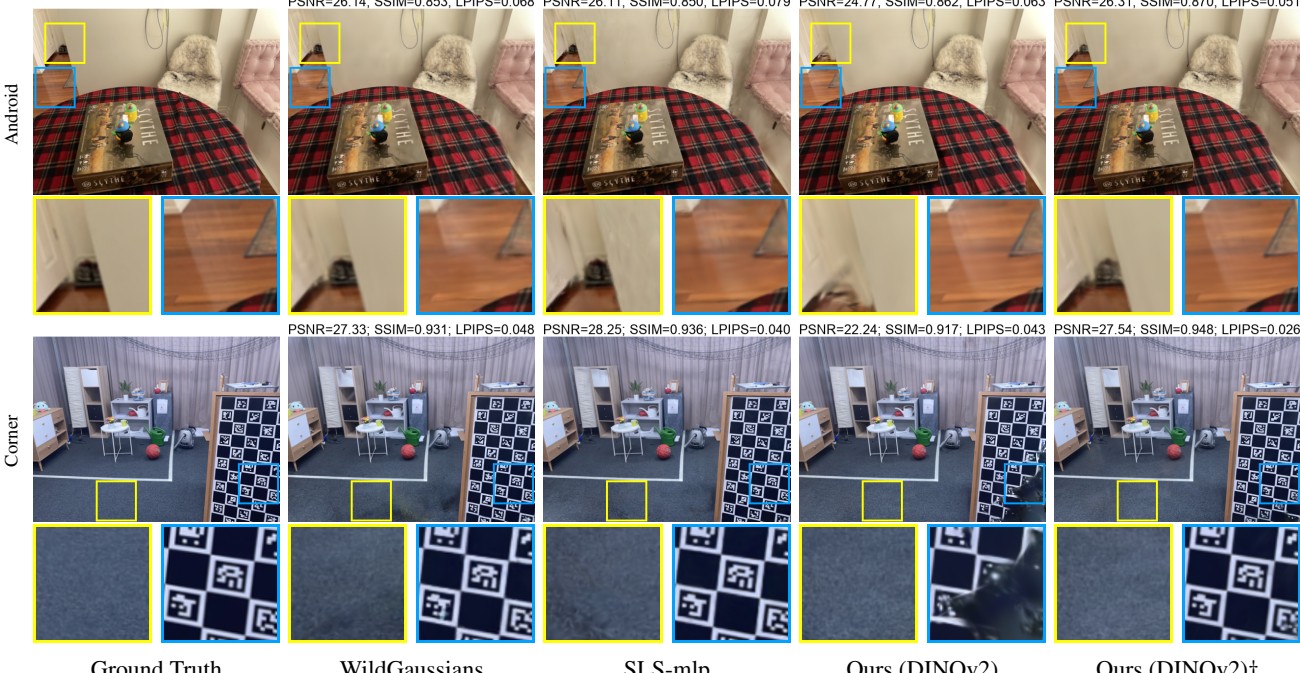

*Figure 12.* **Comparisons on Challenging Cases.** Our method achieves better visual quality in most regions (*e.g.*, walls and floors) with default settings, while it can further capture frequently occluded details (*e.g.*, shoes and plaid patterns) by increasing warm-up steps (†).

*Table 5.* **Extreme Stress Test where Distractors Occupy More Than 50% of the View.** Our method can successfully recover the appearance by avoiding catastrophic failures.

| Method | Avg. PSNR↑ | Avg. SSIM↑ | Avg. LPIPS↓ |
|---|---|---|---|
| 3DGS | 15.93 | .624 | .545 |
| 3DGS w/ GT Masks (Upper Bound) | 28.84 | .992 | .117 |
| **Ours** | 24.85 | .820 | .229 |

2025) uses semantic priors as model input, while WildGaussians (Kulhanek et al., 2024) and T-3DGS (Markin et al., 2024) utilize perceptual error metrics. Compared with them, Ours (ResNet) achieves the smallest GPU memory usage and is only slower than SLS-mlp in average training time.

### B.3. Trade-off in Object Boundaries

As discussed in Sec. 4.1, coarse boundaries are detrimental in 2D segmentation, but the paradigm shifts in 3DGS due to multi-view consistency. If a patch slightly over-masks a distractor, background pixels are merely "unseen" in one view but are effortlessly recovered from others. As shown in Tab. 4, we proved this using manually annotated pixel-level GT masks with different operations during training on the RobustNeRF dataset. Method (3) simulates our coarse strategy, achieving comparable or better quality than baseline (1). Conversely, (4) shows that misclassifying regions causes severe degradation. These prove our argument: **imperfect boundaries do not necessarily hurt reconstruction quality if transient regions are correctly identified**.

### B.4. GMM Assumption with Dominant Distractors

When distractors dominate a scene, they change the proportion of the error distribution but do not flip the relative means of GMM components. Transients remain multi-view inconsistent, always producing higher reconstruction errors than static backgrounds. Therefore, the lower-mean component robustly represents static pixels, entirely independent of component size. To prove this, we simulated extreme pressure tests on RobustNeRF scenes by adding random colored patches occluding more than 50% of every training image. The GMM assumption holds reliably in this stress test: 73.62% of static pixels are correctly assigned to the lower-mean component, and 88.83% of transients to the higher-mean. As shown in Tab. 5, this robust classification prevents catastrophic failure and successfully recovers the 3D geometry.

### B.5. More Results on Android & Corner Scenes

As shown in Figs. 6 and 7 of the main paper, our method outperforms previous methods in most visual metrics across various scenes. However, it brings less improvement in PSNR on Android and Corner scenes compared to others. We ascribe this to a longstanding challenge in distractor-free NeRF/3DGS methods: the inherent ambiguity between infrequently observed static objects and genuinely transient ones. Since transient objects are typically identified based on reconstruction errors arising from their limited presence in the training images, rarely visible static content may be

| Method | Brandenburg Gate | | | Sacre Coeur | | | Trevi Fountain | | | Avg. | | |
|---|---|---|---|---|---|---|---|---|---|---|---|---|
| | PSNR↑ | SSIM↑ | LPIPS↓ | PSNR↑ | SSIM↑ | LPIPS↓ | PSNR↑ | SSIM↑ | LPIPS↓ | PSNR↑ | SSIM↑ | LPIPS↓ |
| 3DGS (Kerbl et al., 2023) | 21.20 | .889 | .131 | 16.73 | .833 | .166 | 17.66 | .729 | .210 | 18.53 | .817 | .169 |
| NeRF-HuGS (Chen et al., 2024) | 24.69 | .890 | .132 | 21.09 | .819 | .169 | 22.63 | .748 | .195 | 22.80 | .819 | .165 |
| WildGaussians (Kulhanek et al., 2024) | 27.77 | .927 | .133 | 22.56 | .859 | .177 | 23.63 | .766 | .228 | 24.65 | .851 | .179 |
| Ours | 20.76 | .879 | .139 | 16.56 | .822 | .177 | 17.31 | .691 | .261 | 18.21 | .797 | .193 |
| Ours w/ Embedding | 27.90 | .932 | .094 | 22.68 | .873 | .126 | 22.35 | .785 | .176 | 24.31 | .863 | .132 |

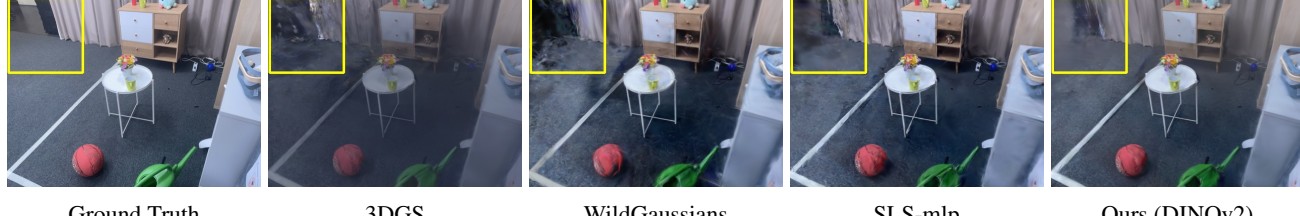

| Training Images | Ground Truth | WildGaussians | Ours | Ours w/ Embed. |

*Figure 13.* **Comparisons on the Phototourism Dataset (Martin-Brualla et al., 2021).** The first , second , and third best values are highlighted. Our method can easily adapt to scenes with appearance variations by assigning embeddings to each image and Gaussian.

| Ground Truth | 3DGS | WildGaussians | SLS-mlp | Ours (DINOv2) |

*Figure 14.* **A Failure Case.** Without prior knowledge from the generative model, 3DGS and its variants cannot correctly predict the scene content that has never been observed before.

mistakenly classified as transient. Nevertheless, our method can mitigate this issue and improve PSNR by straightforward hyperparameter tuning (*i.e.*, the warm-up steps) to preserve more static details (Fig. 12). Please see Fig. 15 for additional qualitative results.

### B.6. Extension to Scenes w/ Appearance Variations

Our method is designed for scenes with stable appearances, but can extend naturally to non-static appearances caused by diverse shooting environments and shooting equipment (*e.g.*, different weather changes or inconsistent white balances). Specifically, we apply a widely adopted paradigm (Chen et al., 2024; Kulhanek et al., 2024; Martin-Brualla et al., 2021) to our method, which assigns learnable appearance embeddings to each image and Gaussian. When rendering a specific image, each Gaussian embedding is combined with the image embedding and processed by a lightweight MLP, thereby scaling and biasing the corresponding Gaussian colors to achieve color changes.

We conduct an experiment on the Phototourism Dataset (Martin-Brualla et al., 2021), which includes three scenes of cultural landmarks with varying appearances and various distractors. As shown in Fig. 13, our extended method can capture appearance variations compared to the original method, and achieves quantitative performance comparable to previous baseline methods, leading in SSIM and LPIPS metrics. In terms of visual quality, our extended

method effectively reduces artifacts caused by appearance variations compared to the original method, resulting in cleaner rendered results.

### C. Limitations and Future Work

As mentioned above, despite the superiority of our method, it still faces a longstanding challenge in distractor-free NeRF/3DGS: reliably distinguishing between transient distractors and infrequently visible static elements. These static elements can be highly reflective objects whose appearance changes drastically under limited views (*e.g.*, the ground reflection in Fig. 12), or objects that are rarely observed (*e.g.*, black shoes in Fig. 12). In such cases, these static regions may be mistakenly treated as transient elements due to insufficient visibility. Therefore, as long as the reconstruction error remains the primary criterion for distinguishing transients, this challenge is likely to persist. Addressing it may require a paradigm shift that could offer substantial benefits to the community. In addition, patch-wise methods may struggle with microscopic objects or extremely thin static structures, because a patch partially occupied by a distractor can over-suppress neighboring static pixels in that view. In practice, most real-world distractors easily exceed common patch sizes (*e.g.*, $16 \times 16$), and genuinely sub-patch distractors act like minor sensor noise that can be inherently smoothed out by 3DGS. Finally, our method inherits the limitations of native 3DGS and its variants, particularly the

inability to reconstruct unobserved scene content (Fig. 14). Future work may address this limitation by incorporating pre-trained generative models.

## D. Additional Qualitative Results

### D.1. Evaluation on Novel View Synthesis

We provide more qualitative results of view synthesis on different datasets. As shown in Fig. 15, existing methods exhibit limitations in managing transient distractors, while our method effectively balances the exclusion of transient distractors and retention of static details, thereby enhancing the rendering quality. Please zoom in for better details.

### D.2. Evaluation on Static Map Generation

We also provide more qualitative results of static maps. As shown in Figs. 16 and 17, existing methods that leverage semantic priors still exhibit limited efficacy in separating transient distractors from static backgrounds. Conversely, our proposed framework can generate more accurate static maps by addressing the inherent limitations of semantics.

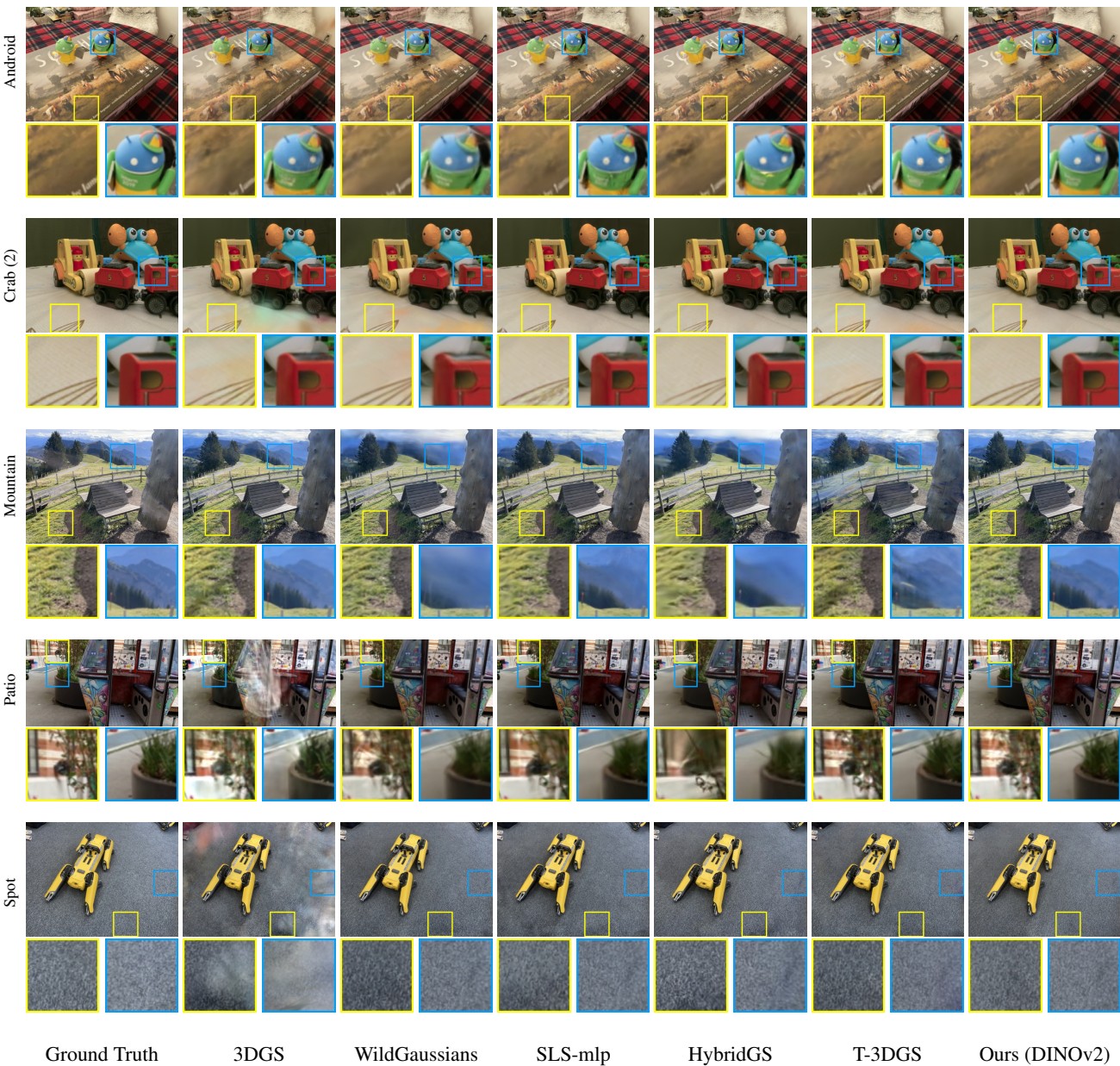

*Figure 15.* **Qualitative Results of View Synthesis on Different Datasets.**

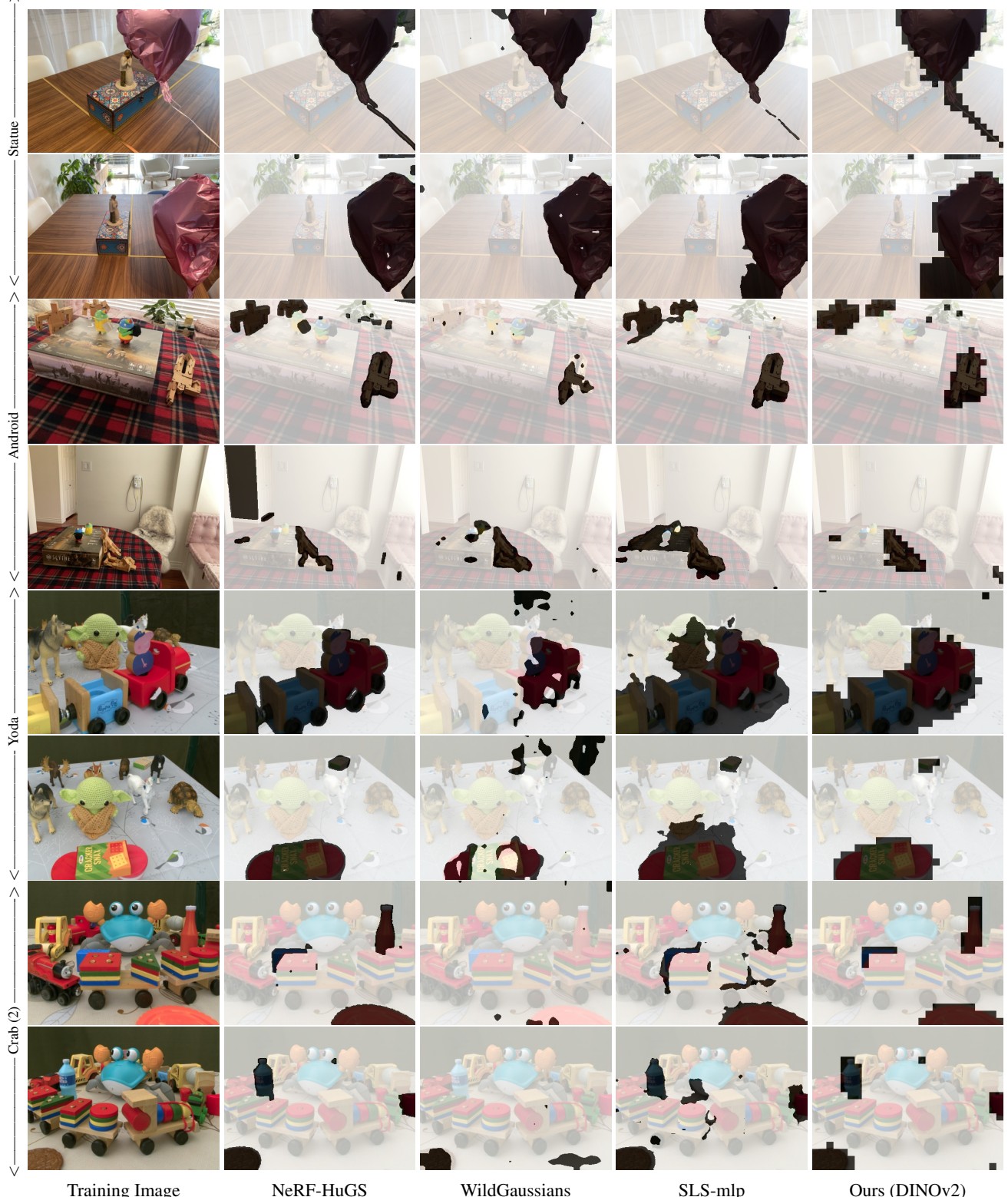

*Figure 16.* **Qualitative Results of Static Map Generation on the RobustNeRF Dataset.**

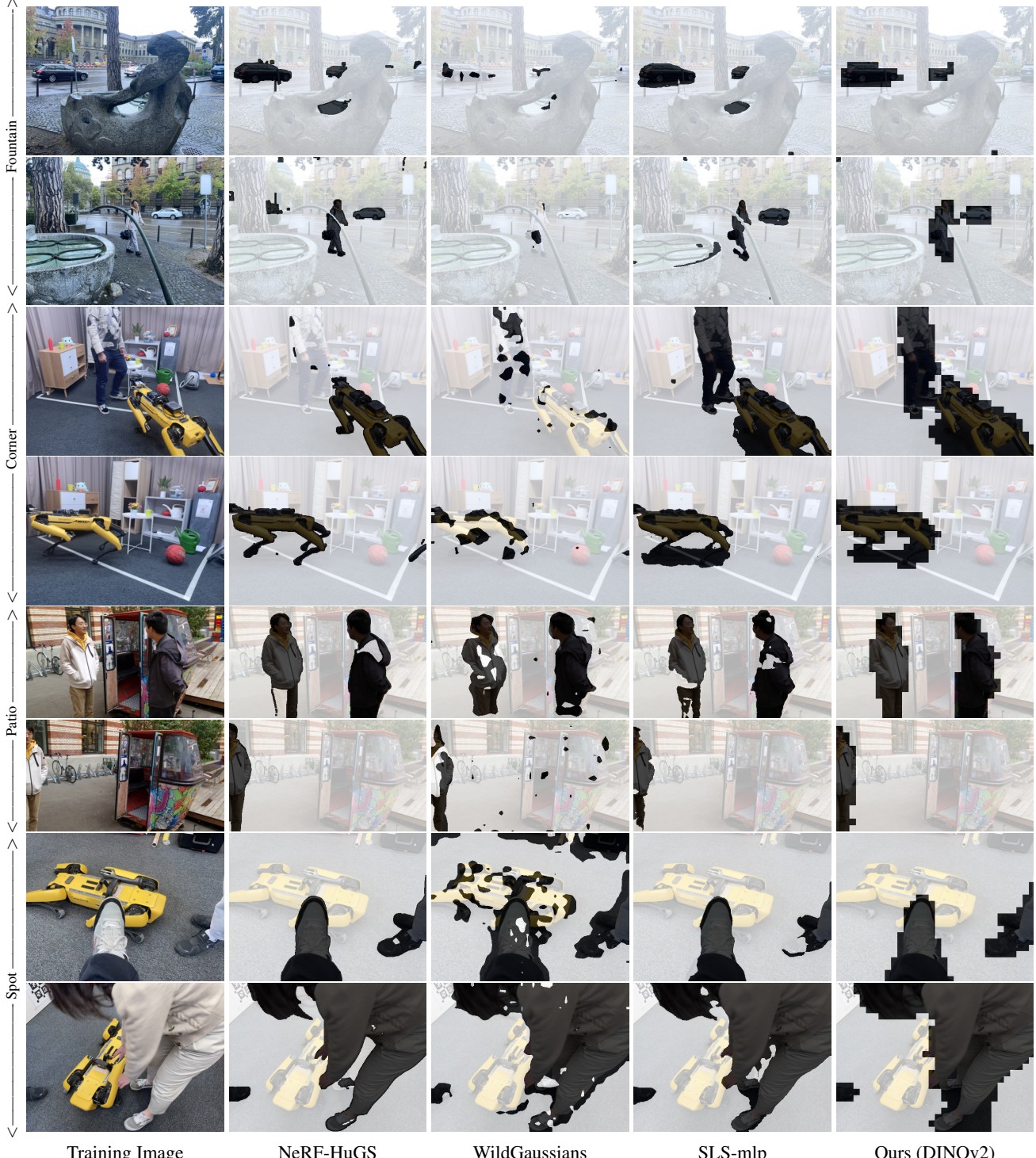

*Figure 17.* **Qualitative Results of Static Map Generation on the On-the-go Dataset.**

