# OpenReview forum: "3DGS-HPC: Distractor-free 3D Gaussian Splatting with Hybrid Patch-wise Classification"
_ICML.cc/2026/Conference — ICML 2026 regular_

### Official Review · Reviewer_ssEZ · 2026-02-15

**Soundness:** 3
**Presentation:** 3
**Significance:** 3
**Originality:** 2
**Overall Recommendation:** 4
**Confidence:** 4

**Summary:**

This paper proposes a novel framework called **3DGS-HPC** to address reconstruction artifacts in 3D Gaussian Splatting caused by transient distractors (e.g., pedestrians and shadows) and the over-reliance on fragile semantic priors. Instead of using complex semantic segmentation pipelines, the method introduces a Hybrid Patch-wise Classification strategy that combines photometric error with perceptual feature similarity to identify and remove distractors. This design improves static background reconstruction quality and visual cleanliness while maintaining efficiency.

**Compliance With Llm Reviewing Policy:**

Affirmed.

**Final Justification:**

Although this work is somewhat lacking in theoretical innovation, it is overall solid and worthwhile. I opt for a weak accept.

**Key Questions For Authors:**

## 1. Patch Boundary Alignment and Potential Over-Suppression

Since patch boundaries do not necessarily align with the true physical edges of objects, could coarse patch masks lead to erroneous removal of static background regions in areas with significant object motion? In particular, when distractors partially occupy a patch, the discretized mask may over-suppress neighboring static content. The authors are encouraged to provide qualitative examples of failure cases, especially around complex object boundaries, to better illustrate the limitations of the patch-based classification mechanism.

## 2. Computational Overhead Compared to Vanilla 3DGS

Compared to standard 3D Gaussian Splatting (3DGS), what is the additional computational overhead introduced by the HPC framework in terms of FLOPs or per-iteration training time? Specifically, the repeated GMM fitting and feature-sequence processing may impose non-trivial computational costs. A detailed efficiency analysis would clarify whether these additional components compromise the efficiency advantage traditionally associated with 3DGS.

## 3. Robustness in Textureless Regions Under Illumination Variations

In textureless regions mentioned in the paper, if L1 photometric error becomes unreliable due to illumination changes or environmental lighting variations, how does the proposed hybrid metric maintain stability? A more detailed explanation of the robustness mechanism in such challenging scenarios would strengthen the method’s validity.

**Limitations:**

yes

**Strengths And Weaknesses:**

# Strengths

## 1. Simplified Pipeline via Patch-wise Partitioning

Unlike approaches that rely on complex pipelines built upon large-scale segmentation models such as Segment Anything (SAM), the proposed method adopts a patch-wise partitioning strategy. This design avoids explicit semantic segmentation modules, resulting in a cleaner framework with improved implementation simplicity and reproducibility.

## 2. Insight into the Semantic–Transience Mismatch

The paper insightfully argues that semantic labels produced by general-purpose vision models (e.g., “person”) do not necessarily correspond to the notion of *transience* in 3D scene reconstruction (e.g., shadows or temporarily occluded static backgrounds). This distinction highlights a conceptual mismatch between semantic categorization and transient-static decomposition, offering a thought-provoking perspective on distractor handling in 3D reconstruction.

## 3. Competitive Quantitative and Qualitative Performance

Quantitative experiments demonstrate that the method achieves first- or second-best performance across multiple metrics in challenging scenes such as *Statue* and *Yoda*. Furthermore, qualitative comparisons show visually cleaner background reconstruction, suggesting improved suppression of transient artifacts.

---

# Weaknesses

## 1. Limitations of Mechanical Non-Overlapping Patch Partitioning

The non-overlapping square patch partitioning is inherently limited in its ability to precisely conform to irregular object boundaries. Although the authors cite prior work suggesting that coarse partitioning has minimal influence on overall reconstruction quality, such grid-based discretization may introduce staircase-like masks or boundary artifacts when handling fine-grained structures or thin object edges.

## 2. Assumption of a Bi-Modal Gaussian Error Distribution

The method assumes that the reconstruction error follows a bi-modal Gaussian distribution. However, in scenes with extreme lighting variations, reflections, or complex shadows, background regions themselves may exhibit multi-modal characteristics. Under such conditions, the convergence stability and classification reliability of the Gaussian Mixture Model (GMM) could be compromised. The paper does not provide robustness analysis under such challenging distributions.

## 3. Lack of Fundamental Justification for Patch-Based Transience Modeling

The paper does not fundamentally explain why patch-wise partitioning would more effectively characterize *transience* compared to fine-grained semantic segmentation. If the benefit primarily arises from the local smoothness induced by patch aggregation, the motivation appears largely heuristic rather than grounded in a deeper theoretical or principled analysis. A more rigorous justification would strengthen the scientific contribution.

## 4. Absence of Theoretical Analysis on Patch Discretization Error

The image is divided into non-overlapping square patches, while distractor boundaries in real-world scenes may take arbitrary shapes. The paper lacks a theoretical analysis of what may be termed *patch discretization error*. For instance, when a distractor occupies only 10% of a patch area, how does the discriminative power of the proposed metric degrade? Without quantifying such boundary-induced effects, the reliability limits of the patch-based decision mechanism remain unclear.

---

> ### Author Rebuttal · Authors · 2026-03-31
>
> We thank Reviewer ssEZ for the positive evaluation and recognizing our insights into the semantic-transience mismatch.
>
> **W1, W4 & KQ1: Patch Discretization, Boundary Artifacts, and Over-Suppression**
>
> We agree non-overlapping patches cause spatial discretization errors. Masking a patch with 10% transient content indeed locally "over-suppresses" the 90% static background in that view. However, 3DGS optimization presents an asymmetry in risk: **local over-suppression of background is highly tolerable, whereas under-suppression of distractors is catastrophic**. Due to multi-view consistency, background pixels masked in one view are effortlessly recovered from other views. Conversely, under-suppressing distractors leaves permanent 3D floaters. Thus, trading redundant background observations for guaranteed distractor removal is highly beneficial.
>
> To quantify this error, we simulated coarse boundaries using pixel-level GT masks (**see response to Reviewer yTAK W2**). Results prove that converting perfect masks into coarse patches doesn’t degrade quality; it yields comparable/better performance by aggressively filtering boundary noise. We will add qualitative failure cases to the appendix, illustrating limits like over-suppressing extremely thin, rarely-observed static structures.
>
> **W2 & KQ3: GMM Assumption and Robustness under Illumination Variations**
>
> We agree that raw L1 photometric error fluctuates and becomes multi-modal under severe lighting variations. Hence, we designed the **hybrid metric** (Sec. 4.2) and integrated **appearance embeddings** for extreme cases (Sec. B.5).
>
> Specifically, the final static map is the intersection of both photometric and perceptual masks. The L1 GMM provides a foundational classification mask and estimates the static proportion to dynamically threshold the perceptual metric. Ablations (Sec. 5.3, Fig. 9\) show this intersection ensures a rigorous "double-check": the L1 proportion prevents perceptual semantic fragility from destroying valid backgrounds, while the perceptual mask filters distractors confusing raw L1.
>
> Furthermore, for scenes with extreme illumination or weather variations where L1 might be heavily distorted, integrating appearance embeddings absorbs exposure changes (Sec. B.5, Fig. 13). This stabilizes the underlying L1 distribution, allowing our hybrid metric to handle complex lighting and outperform baselines.
>
> **W3: Fundamental Justification for Patch-Based Modeling**
>
> Replacing semantic segmentation with patches is rooted in the "Semantic Mismatch" (Sec. 4.1), arising from the fundamental conflict between **object-centric priors** and **state-centric (static/transient) reality**.
>
> Semantic models group pixels by object identity. However, transience often alters a region's physical state without changing its semantic category. For example, a semantic model may group both the static floor and a transient cast shadow together under the same "ground" label. Forcing distractor masks to align with these semantic boundaries introduces a flawed inductive bias that inherently misses such intra-object dynamic phenomena.
>
> Our patch-wise strategy deliberately discards this object-level prior. Instead, it relies purely on **local spatial consistency**: the principled physical assumption that neighboring pixels share the same transient/static state. By completely decoupling the masking process from object identity, patches can flexibly isolate a transient shadow from a static floor based on their actual multi-view consistency errors. Our method is not designed to be more "fine-grained" than semantics; rather, it is theoretically more **unbiased**, accurately capturing state changes that semantic methods fundamentally overlook (Figs. 3 & 8).
>
> **KQ2: Computational Overhead Compared to Vanilla 3DGS**
>
> The computational overhead introduced by HPC is minimal, preserving 3DGS's training efficiency:
> 1. **Infrequent Feature Extraction & Lightweight Models:** We do not extract deep features at every step. As detailed in Secs. 5.1 & A, masking is entirely skipped during the 500-step warm-up, and features are only extracted every 100 iterations via lightweight encoders.
> 2. **Accelerated GMM Fitting:** The patch-wise partition significantly reduces the classification computational cost (Sec. B.1, Tab. 2). By aggregating pixels into patches, the 1D GMM fits a highly compact sequence of only patch-level errors. This fitting takes mere milliseconds.
>
> Empirically, **Tab. 3 (Sec. B.2)** shows that our ResNet variant requires an average of only **13.29 mins/scene**. This is significantly faster than semantic-heavy baselines like WildGaussians (24.10 mins) and T-3DGS (26.75 mins), and is highly competitive with native 3DGS.

---

> > ### Author Rebuttal · Reviewer_ssEZ · 2026-04-02
> >
> > Good luck

---

### Official Review · Reviewer_MbKY · 2026-02-17

**Soundness:** 3
**Presentation:** 3
**Significance:** 3
**Originality:** 3
**Overall Recommendation:** 4
**Confidence:** 4

**Summary:**

This paper proposes a framework called 3DGS-HPC for robust 3D Gaussian splatting in scenes with transient distractors. The authors note that existing distractor removal methods rely on external semantic priors. These methods suffer from semantic mismatch and vulnerability to appearance perturbations. To address these issues, 3DGS-HPC introduces two main components. The first component is a patch based classification strategy. This strategy utilizes local spatial consistency instead of external semantic grouping. The second component is a hybrid classification metric. This metric combines photometric and perceptual error cues. Specifically, the method utilizes a Gaussian mixture model based on photometric errors. This model estimates the global static pixel ratio to guide percentile thresholding based on perceptual errors. Evaluations on RobustNeRF and On the go datasets demonstrate the effectiveness of this approach. The method significantly improves novel view synthesis quality and static mask accuracy compared to state of the art baselines. It also maintains competitive training efficiency.

**Compliance With Llm Reviewing Policy:**

Affirmed.

**Final Justification:**

After carefully evaluating the authors' rebuttal, I have decided to increase my score from 3 to 4. The authors validated the robustness of the GMM assumption through extreme stress tests and clarified the conceptual differences with DGGS, effectively addressing my concerns regarding stability and originality. While the use of semantic features warrants further optimization, the method's ability to achieve robust transient removal without heavy external segmentation models represents a significant contribution.

**Key Questions For Authors:**

Please see weakness. I will consult the assessments provided by the other reviewers. I will consider increasing my score if my current concerns are completely resolved.

**Limitations:**

Yes

**Strengths And Weaknesses:**

**Strengths:**

- The paper proposes a patch-wise classification strategy to identify transient distractors, effectively bypassing the need for computationally heavy and semantically mismatched external segmentation models by relying on local spatial consistency.

- Extensive evaluation is conducted on real-world datasets like RobustNeRF and On-the-go to verify the performance.

**Weakness**：
- Overall, while the patch-wise approach is neat, the paper misses a critical conceptual discussion regarding contemporary work. DGGS (Bao et al., 2024) is cited merely as a pixel classification method, but it actually relies heavily on Entity Segmentation for mask refinement. Since the DGGS code isn't open-source, an empirical comparison isn't expected, but how does the proposed segmentation-free approach theoretically contrast with DGGS's entity-reliant masking in the exact same domain?
- How to theoretically ensure that rarely observed static elements (e.g., highly reflective surfaces or heavily occluded backgrounds) are not mistakenly filtered out? The model relies entirely on reconstruction errors , which inherently penalizes sparse static observations just like transient objects.
- The model performance seems overly sensitive to hyperparameters, specifically the warm-up steps and patch size. In Figure 12, performance drops sharply if the warm-up exceeds 1,000 steps, indicating that the model easily overfits to distractors. How does the framework handle real-world deployment if it requires tedious per-scene manual tuning for these steps?
- How is the GMM two-component assumption guaranteed to hold? The method assumes the distribution with the lower mean always represents static pixels. What happens if transient distractors dominate a scene (e.g., occupying >50% of the view)? There is no formal analysis or failure case shown for when this core assumption breaks.
- There’s a clear contradiction in Section 4.2 : the authors criticize the "semantic fragility" of foundation models, yet their pipeline still relies entirely on frozen DINOv2/VGG features for the perceptual metric. Doesn't this mean the model is just inheriting the very object-centric biases it complains about? Also, periodically extracting and aligning these heavy deep features during the 3DGS training loop has to be computationally expensive, but the paper completely glosses over this overhead.

---

> ### Author Rebuttal · Authors · 2026-03-31
>
> We thank Reviewer MbKY for the constructive feedback.
>
> **W1: Conceptual Contrast with DGGS (Bao et al., 2024)**
>
> We will explicitly discuss DGGS in our revision. We categorized these works as "pixel classification" because their paradigm is identical: per-pixel prediction followed by segmentation-based refinement. DGGS relies on **entity-centric priors**, assuming object boundaries align perfectly with static/transient boundaries. As analyzed in Sec. 4.1, this object-centric prior suffers from semantic mismatches and struggles with non-entity transient phenomena (**see response to Reviewer ssEZ W3**).
>
> Conversely, our **segmentation-free patching** intentionally discards the "entity" concept. By relying purely on local spatial consistency, we structurally bypass these semantic mismatches. Furthermore, we directly classify patches instead of using the computationally heavy "pixel-classification then mask-refinement" pipeline, significantly accelerating training while maintaining robustness (Sec. B.1).
>
> **W2: Misclassification of Rarely Observed Static Elements**
>
> We have analyzed this vulnerability in **Secs. B.3 & C**, acknowledging it as **a long-standing challenge inherent to the entire field**. Theoretically, it is impossible to perfectly guarantee zero misclassification for extremely sparse observations without novel external priors. However, our framework significantly mitigates this vulnerability through two mechanisms:
> 1. **Robustness of Hybrid Perceptual Metrics**. Pure photometric error heavily penalizes view-dependent specular reflections. By contrast, our hybrid metric integrates perceptual features to capture structural semantics that are far more invariant to lighting/color shifts, preventing reflective static surfaces from being falsely flagged.
> 2. **Tolerance of Patch-wise Aggregation**. Naive pixel-wise methods often permanently delete sparse high-error pixels early. Our spatial patching smooths errors, granting sparse static pixels a buffer to survive the initial thresholding until 3DGS optimization converges.
>
> **W3: Hyperparameter Sensitivity**
>
> We clarify that **we didn't perform any per-scene manual tuning**. We strictly used fixed default settings across all scenes, powerfully outperforming baselines out-of-the-box. The performance drop after 1,000 steps (Fig. 12\) does not indicate hyperparameter fragility, but reflects the fast convergence of native 3DGS. If masking is delayed beyond 1,000 steps, 3DGS permanently "memorizes" transients as 3D geometry (overfitting). Thus, our default of 500 steps is not a finely-tuned scene-specific value, but a universal, physically grounded constant to initiate filtering before memorization occurs.
>
> **W4: GMM Assumption with Dominant Distractors (>50%)**
>
> When distractors dominate, **it changes the proportion of the error distribution, but does not flip the relative means of GMM components**. Transients remain multi-view inconsistent, always producing higher reconstruction errors than static backgrounds. Therefore, the lower-mean component robustly represents static pixels, entirely independent of component size.
>
> To prove this, we simulated extreme pressure tests on RobustNeRF scenes by adding random colored patches occluding **>50% of every training image**. The GMM assumption holds perfectly: **73.62%** of static pixels are correctly assigned to the lower-mean component, and **88.83%** of transients to the higher-mean. As shown below, this robust classification prevents catastrophic failure and successfully recovers the 3D geometry. We will add this extreme-case stress test to our supplementary materials.
>
> | Method (Trained under >50% occlusion) | Avg. PSNR | Avg. SSIM | Avg. LPIPS |
> | :---- | :---: | :---: | :---: |
> | 3DGS | 15.93 | .624 | .545 |
> | **Ours** | 24.85 | .820 | .229 |
> | 3DGS w/ GT Masks (Upper Bound) | 28.84 | .992 | .117 |
>
> **W5: Semantic Contradiction & Computational Overhead**
>
> We clarify that there is no contradiction, but rather a distinction in how foundation models are utilized (**see response to Reviewer yTAK KQ2**). We oppose using semantics for *intra-image grouping*, which imposes object-centric mismatches. Conversely, we utilize semantics purely as *cross-image metrics* to resist color ambiguities. To prevent inheriting their "semantic fragility", our pipeline strategically calibrates the perceptual threshold using the photometric GMM. Ablations show our calibrated usage performs better than naively relying on semantic priors.
>
> We have detailed the computational efficiency in **Secs. B.1 & B.2**, which is achieved through infrequent extractions (e.g., every 100 steps) and lightweight models (e.g., DINOv2 ViT-S). We also completely bypass the need for per-pixel classification or heavy instance segmentation by directly classifying patches. Tab. 3 shows our method averages **13.29 mins/scene**, significantly faster than most baselines.

---

> > ### Author Rebuttal · Reviewer_MbKY · 2026-04-01
> >
> > Thank you to the authors for their thorough and constructive rebuttal. I am pleased to raise my score to reflect the solid contribution of this paper. All my concerns have been fully addressed.

---

### Official Review · Reviewer_gSou · 2026-03-12

**Soundness:** 3
**Presentation:** 3
**Significance:** 4
**Originality:** 4
**Overall Recommendation:** 5
**Confidence:** 3

**Summary:**

This paper introduces 3DGS-HPC, a novel framework designed to achieve robust, distractor-free 3D Gaussian Splatting in real-world environments by effectively filtering out transient elements like moving objects and shadows. Recognizing that existing semantic-based removal methods suffer from semantic mismatch and fragility to minor appearance perturbations , the authors propose two complementary strategies to accurately generate binary masks for training. First, a patch-wise classification approach leverages local spatial consistency rather than external vision foundation models, grouping pixels into regular patches to improve classification context, robustness, and computational efficiency. Second, a hybrid classification metric adaptively fuses both photometric and perceptual cues, utilizing the global transient pixel proportion estimated from traditional photometric errors to guide the thresholding of perceptual metrics, ensuring a much more reliable separation of static and transient regions. Extensive evaluations on benchmark datasets demonstrate that 3DGS-HPC consistently outperforms state-of-the-art methods in preserving high-fidelity static backgrounds while successfully mitigating complex dynamic distractors.

**Compliance With Llm Reviewing Policy:**

Affirmed.

**Final Justification:**

The experiments solve my concern and I tend to accept.

**Key Questions For Authors:**

No

**Limitations:**

Yes.

**Strengths And Weaknesses:**

### **Strengths**

* **Excellent Clarity and Presentation:** The paper is exceptionally well-written and logically structured. The authors clearly articulate the specific bottlenecks of current semantic-based 3DGS methods—specifically "semantic mismatch" and "semantic fragility"—making the motivation behind their work highly compelling and easy to follow.
* **Methodological Novelty:** The proposed Hybrid Patch-wise Classification (HPC) framework introduces a refreshingly novel and pragmatic perspective. Rather than relying on increasingly heavy vision foundation models for complex pixel-wise segmentation, the approach elegantly leverages local spatial consistency via a simple patch-wise strategy and a hybrid error metric to adaptively separate static and transient elements.


* **Compelling Empirical Results:** The method delivers strong, state-of-the-art performance. Comprehensive evaluations on benchmarks like the RobustNeRF and On-the-go datasets demonstrate that HPC significantly improves quantitative metrics (e.g., PSNR and SSIM) while visually preserving high-fidelity static details and effectively removing complex distractors. It also proves to be highly memory-efficient and fast to train.



### **Weaknesses**

* **No Obvious Weaknesses:** This is a remarkably solid paper with no glaring technical or structural flaws. The experimental design is thorough, the ablation studies convincingly justify the specific architectural choices, and the authors are transparent about the inherent, field-wide challenges that still remain (such as the ambiguity between moving objects and rarely observed static features).

---

> ### Author Rebuttal · Authors · 2026-03-31
>
> We sincerely thank the reviewer for the highly positive evaluation and for recognizing the novelty, clarity, and strong empirical results of our HPC framework. We are glad you found the paper exceptionally well-written and our patch-wise perspective refreshingly pragmatic.

---

> > ### Author Rebuttal · Reviewer_gSou · 2026-04-01
> >
> > I keep my rating.

---

### Official Review · Reviewer_yTAK · 2026-03-12

**Soundness:** 3
**Presentation:** 3
**Significance:** 3
**Originality:** 2
**Overall Recommendation:** 4
**Confidence:** 4

**Summary:**

This paper aims to identify binary static/transient masks from the discrepancies of training and rendered images, thereby reducing the influence of transient distractors during reconstruction. The main contributions are to improve the region (for further classification) definition and the classification metrics. The paper proposes a patch-based partitioning to avoid semantic mismatches caused by external semantic priors, together with a hybrid metric that combines photometric and perceptual cues. Extensive experiments are conducted on two standard datasets and against a wide range of baselines, demonstrating superior performance, proving its simplicity and effectiveness.

**Compliance With Llm Reviewing Policy:**

Affirmed.

**Final Justification:**

I have updated my score slightly after the carefully written rebuttal.

**Key Questions For Authors:**

1. Quantitative results show significant improvement over previous work. However, the gains observed in ablation appear to be relatively minor, making it difficult to assess whether the two components are the main factors contributing to the final improvement, or merely beneficial but not decisive design choices. The author's clarification and analysis on the modest performance changes in the ablation study would be helpful.

2. Could you refine the positioning of the paper with respect to the semantic priors?  Specifically, some passages suggest that this method avoids relying on external semantics and criticize the visual foundation model for not being designed for static/transient classification tasks, while the classification metrics still rely on the perceptual features of the base model. This is not a fundamental contradiction, as the paper argues mainly against using semantic priors alone for region grouping, rather than against the use of semantic features as one of the cues for region classification. However, the current wording is somewhat too strong and confusing. Clarifying this distinction will make the paper's arguments more precise and more consistent with the actual practice of this method.

**Limitations:**

The authors should further discuss the impact of inaccurate object boundaries caused by patch-based partitioning, and the limitations that may require patch size tuning for cluttered scenes with mixed objects of varying sizes.

**Strengths And Weaknesses:**

Strengths:
- The paper clearly defines its scope to differentiate with explicit dynamic modeling. The main paper discusses the static map estimation problem in detail. And the supplementary further discusses the extension to illumination-changed settings. In general, the scope is clear and the discussion is comprehensive.
- The paper is well structured. The motivation and two challenges, together with the two proposed components to address these challenges, are presented clearly and are easy to follow.
- The method is evaluated on standard datasets and includes comparisons with 9 baselines.
- The paper provides informative visualizations. The qualitative figures, e.g., Figure 3-5 help support the paper's claim for motivation.

Major Weakness
- The core contributions appear somewhat incremental. It mainly alternates the design choice on a two-step general "region proposal - region classification" pipeline. The two cues fused and the adaptive control GMM strategy are from previous work.
- The trade-off in precise object boundaries is not fully discussed. The paper argues that imperfect boundaries do not necessarily hurt reconstruction quality if transient regions are correctly identified, which may require more proof. The related work cited for this argument, however, admits that the patch-based method results in inconsistent predictions for small objects. Including supplementary ablation on patch sizes in the main paper can help complement this discussion. However, this also proves that patch size is a parameter with an optimal setting for each scene, which to some extent harms the generalizability of the proposed method.


Minor Weakness
- There are scenes in which the proposed methods perform worse than existing methods. The failure case is not discussed in the main paper but analyzed in the supplementary materials. The main paper would benefit from briefly summarizing representative failure cases as well.
- The comparisons are sufficient, although the paper could be further strengthened by comparing against the most recent distractor-free 3DGS methods, e.g. from RobustSplat from ICCV 2025.

---

> ### Author Rebuttal · Authors · 2026-03-31
>
> We thank Reviewer yTAK for the constructive feedback.
>
> **W1: Novelty & Contribution**
>
> Our contribution lies not in swapping components within the "region proposal-classification" pipeline, but in new insights and a practical understanding of the fundamental limitations of current methods:
> 1. **Flawed Semantic Partitioning.** Recent methods use semantic priors for pixel-wise classification, introducing systematic mismatches when distinguishing static/transient regions. Our insight is that task-specific semantic grouping is actually harmful here, so we revert to a semantics-free patch-wise partition grounded in local spatial consistency.
> 2. **Fragility of Perceptual Metrics.** Recent methods use the perceptual error between rendered and training images. We observe that it may fail due to the semantic fragility of foundation models in textureless areas.
> 3. **Non-trivial Classification Strategy.** We integrate cues complementarily: using GMM classification on photometric error to dynamically estimate global static proportions, calibrating perceptual error thresholds. This adaptive scheme mitigates individual cue limitations.
>
> These novel insights have not been discussed before, leading to a simple yet effective method. We hope the novelty of our design philosophy and analytical contributions would be considered, which offer value beyond just introducing a new module.
>
> **W2: Trade-off in Object Boundaries**
>
> We will move the patch size ablation to the main text and address concerns below:
> * **Boundary Trade-off Proof.** Coarse boundaries are detrimental in 2D segmentation, but the paradigm shifts in 3DGS due to the multi-view consistency. If a patch slightly over-masks a distractor, background pixels are merely "unseen" in one view but effortlessly recovered from others. We proved this using manually annotated pixel-level GT masks on the RobustNeRF dataset. Method (3) simulates our coarse strategy, achieving comparable/better quality than baseline (1). Conversely, (4) shows that misclassifying regions causes severe degradation. These prove our argument: **imperfect boundaries do not necessarily hurt reconstruction quality if transient regions are correctly identified.**
>
> | GT Masks | Avg. PSNR | Avg. SSIM | Avg. LPIPS |
> | :- | :-: | :-: | :-: |
> | 1) Unchanged | 29.66 | .906 | .074 |
> | 2) Dilate 5 pixels | 30.01 | .908 | .074 |
> | 3) Convert to 16x16 patches | 29.93 | .906 | .077 |
> | 4) Misclassify 10% of (3) | 29.35 | .901 | .081 |
>
> * **Small Objects.** We agree that patch methods struggle with microscopic objects, but real-world distractors easily exceed 16x16 patches. Also, genuinely sub-patch distractors act like minor sensor noise, inherently smoothed out by 3DGS. We will discuss this limitation in the revision.
> * **Generalizability.** We didn't perform per-scene tuning. We strictly used a default patch size of 16 across all scenes for comparisons. While Fig. 11 implies that the optimal size might vary per scene, this default consistently outperforms baselines, proving our framework works powerfully out-of-the-box.
>
> **W3 & W4: Failure Cases & RobustSplat Comparison**
>
> We will summarize failure cases in the main text. Also, our method quantitatively outperforms the recent RobustSplat (ICCV 2025):
>
> |RobustNeRF/On-the-go|Avg. PSNR|Avg. SSIM|Avg. LPIPS|
> | :- | :-: | :-: | :-: |
> | RobustSplat | 29.36 / 23.22 | .895 / .818 | .135 / .149 |
> | **Ours** | **29.81 / 23.86** | **.913 / .849** | **.074 / .106** |
>
> **KQ1: Baseline vs. Ablation Gains**
>
> The discrepancy arises because **our ablation baselines already integrate parts of our novel design**, intrinsically avoiding prior catastrophic failures. We study key components by isolating them in a controlled setup (i.e., retaining the hybrid metric in Fig. 8 and retaining patch-wise partition in Fig. 9). Thus, even our weakest baselines (a) outperform most external methods (Figs. 6 & 7). Furthermore, transitioning to our full pipeline yields a substantial **\~1 dB PSNR gain** on On-the-go in both settings, visually corresponding to accurate distractor localization.
>
> **KQ2: Positioning with Respect to Semantic Priors**
>
> We will soften our wording to clarify our precise stance. There is no fundamental contradiction; our work is a critical reflection and improved utilization of semantic priors:
> 1. **Intra-image Partitioning (What we oppose)**: Existing methods use semantics to cluster pixels within an image. We criticize this because semantic boundaries inherently mismatch static/transient boundaries, and replace this with a semantics-free patch strategy.
> 2. **Cross-image Metrics (What we refine)**: Existing methods use semantics to measure perceptual differences. While reasonable for resisting color ambiguities, we criticize its semantic fragility in specific areas. We introduce a dynamic thresholding strategy to actively correct these failures.
>
> Ablations show naively relying on semantic priors for either step fails, while our calibrated usage achieves SOTA performance.

---

> > ### Author Rebuttal · Reviewer_yTAK · 2026-04-01
> >
> > Thank you for the provided rebuttal that carefully addresses my questions. I will slightly increase my score.

---

### Decision · Program_Chairs · 2026-04-30

**Decision:**

Accept (regular)

**Comment:**

This paper received mixed borderline recommendations during the preliminary reviews. The reviewers appreciated that the paper was clearly articulated, scoped and structured, that the approach is simple but sensible and does not rely on external heavy-weight models, that the results are strong, and that the limitations are transparent. Several weaknesses were raised by the reviewers, many in common, including that the contributions are incremental and somewhat lacking in theoretical innovation, that some parameters may inhibit generalizability, that the ablation study does not indicate a decisive benefit, that the two-component GMM assumption may not be valid, and that sparsely-observed static elements may be incorrectly filtered. Most of these concerns were addressed very well in the author response with both empirical evidence and clear explanations, including considerable validation of the GMM assumption and patch size (in)sensitivity, and an explication and extension of the ablation results. It is clear that all reviewers found the rebuttals compelling and helpful. The AC concurs and recommends that the authors integrate these insightful responses in their revision. The resulting recommendations were all positive, with one strongly arguing for acceptance, and the others being satisfied that the contribution is solid and worth communicating. Given the quality of the reviews, response and engagement, the AC sees no reason to override the consensus of the reviewers and is satisfied by the technical merit of the paper, the insights it shares, and its potential to make an impact on the field.